# Activation mechanism of the class D fungal GPCR dimer Ste2

Vaithish Velazhahan[1], Ning Ma[2], Nagarajan Vaidehi[2] & Christopher G. Tate[1 ✉]

The fungal class D1 G-protein-coupled receptor (GPCR) Ste2 has a different arrangement of transmembrane helices compared with mammalian GPCRs and a distinct mode of coupling to the heterotrimeric G protein Gpa1–Ste2–Ste18[1]. In addition, Ste2 lacks conserved sequence motifs such as DRY, PIF and NPXXY, which are associated with the activation of class A GPCRs[2]. This suggested that the activation mechanism of Ste2 may also differ. Here we determined structures of *Saccharomyces cerevisiae* Ste2 in the absence of G protein in two different conformations bound to the native agonist α-factor, bound to an antagonist and without ligand. These structures revealed that Ste2 is indeed activated differently from other GPCRs. In the inactive state, the cytoplasmic end of transmembrane helix H7 is unstructured and packs between helices H1–H6, blocking the G protein coupling site. Agonist binding results in the outward movement of the extracellular ends of H6 and H7 by 6 Å. On the intracellular surface, the G protein coupling site is formed by a 20 Å outward movement of the unstructured region in H7 that unblocks the site, and a 12 Å inward movement of H6. This is a distinct mechanism in GPCRs, in which the movement of H6 and H7 upon agonist binding facilitates G protein coupling.

The active-state structure of the class D fungal GPCR Ste2 coupled to the heterotrimeric G protein Gpa1–Ste4–Ste18 differs from that of other GPCR classes[1]. Ste2 is a homodimer that can couple to two G proteins simultaneously and its dimer interface is formed by the N terminus, extracellular loop 1 (ECL1) and transmembrane helices H1 and H7. The arrangement of transmembrane helices in Ste2 differs from receptors in other classes, with the intracellular end of H4 shifted approximately 20 Å and the α5 helix of the G protein fitting into a ledge formed by the intracellular ends of H3, H4 and H5. In addition, H5 and H6 are not curved or kinked outwards from the receptor as observed in both class A and class B GPCR G-protein-coupled states. The combination of these features suggested that the activation mechanism of Ste2 may be different from other GPCRs. To study this further we determined structures by single-particle cryo-electron microscopy (cryo-EM) of *S. cerevisiae* Ste2 either in the ligand-free state or bound either to an antagonist (Ste2–Ant) or an agonist (Ste2–Ag), and compared them with the G-protein-coupled state[1] (Ste2–Ag–G).

The inactive-state structure Ste2–Ant was determined using the antagonist[3] [desTrp[1]Ala[3]Nle[12]]α-factor (HALQLKPGQP(Nle)Y), hereafter referred to as 'antagonist'. The active-state structures of Ste2–Ag were determined bound to the native agonist, pheromone α-factor (WHWLQLKPGQPMY). The antagonist is a mutated form of α-factor with Trp1 deleted and the W3A mutation added[4]; conversion of Met12 to Nle does not affect signalling but makes the peptide more resistant to oxidation. Expression and purification of wild-type Ste2 in the ligand-bound states (Extended Data Fig. 1a, b) was similar to that described previously[1] (Methods). We devised a method that we call pre-stabilization of a GPCR by weak association (PSGWAY) to purify Ste2–LF (Methods,

Extended Data Fig. 1c) because of its inherent instability. The structures (Fig. 1a–d) were determined by single particle cryo-EM (Extended Data Figs. 1d–g, 2) to overall resolutions of 3.1 Å (ligand-free) and 2.7 Å (Ste2–Ant), with two separate structures of Ste2–Ag determined to resolutions of 3.5 Å (Extended Data Figs. 3a–d, Extended Data Table 1). Two-fold symmetry (C2) was imposed on the homodimer during the last stages of cryo-EM map determination, as this improved the overall map resolution (Methods, Extended Data Fig. 2). The structures all exhibited clear density for the majority of the side chains (Extended Data Fig. 4a–d, g) and, where present, the ligands (Fig. 1e–g, Extended Data Fig. 4f). All of the structures contained densities on the periphery of the transmembrane regions of Ste2 that were attributed to putative sterols (Fig. 1a–d, Extended Data Fig. 4g). These were either unmodelled or, where the density was sufficiently strong, assigned putatively as cholesterol hemisuccinate (CHS) as this was present in vast molar excess throughout receptor purification; we cannot exclude the possibility that the densities may represent other sterols.

## Ligand-free and antagonist-bound states

The overall structures of ligand-free Ste2 and Ste2–Ant are almost identical (root mean square deviation (r.m.s.d.) 0.5 Å), with the exception of the extracellular end of H5, which is kinked outwards in ligand-free Ste2 and forms a regular helix in Ste2–Ant (Extended Data Fig. 5b). This region is also highly flexible, as observed in the 3D variability analysis of ligand-free Ste2 (Extended Data Fig. 6a), consistent with weak density for some side chains (for example, Asp201[5x25]) and alternate positions for other side chains (F204[5x28] and Lys202[5x26]) (Extended Data Fig. 3e–j)

[1]MRC Laboratory of Molecular Biology, Francis Crick Avenue, Cambridge, UK. [2]Department of Computational and Quantitative Medicine, Beckman Research Institute of the City of Hope, Duarte, CA, USA. ✉e-mail: cgt@mrc-lmb.cam.ac.uk

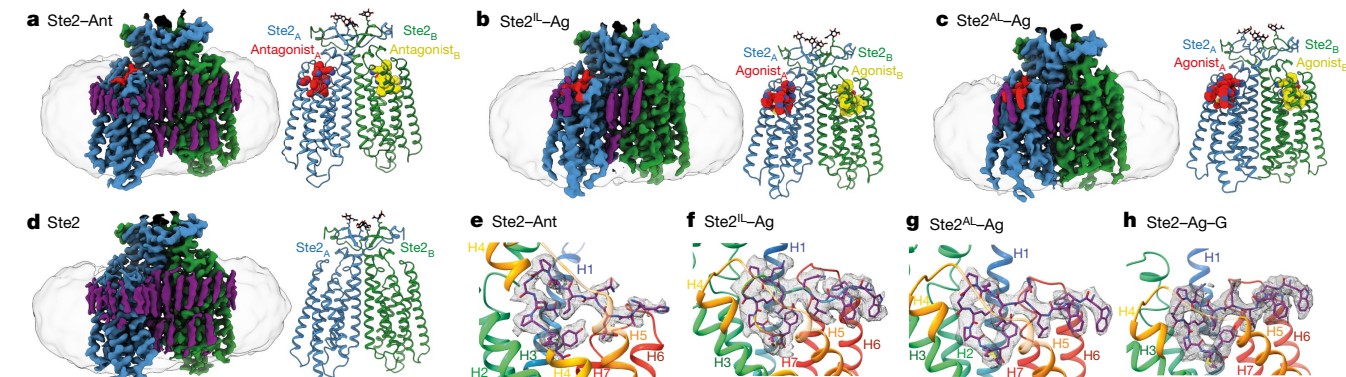

**Fig. 1 | Cryo-EM structures of different conformational states of Ste2.**
**a**–**d**, Cryo-EM density map (left) and structure model (right) for Ste2–Ant (**a**), Ste2[IL]–Ag (**b**), Ste2[AL]–Ag (**c**) and ligand-free Ste2 (**d**) (Ste2$_A$ (blue), Ste2$_B$ (green), CHS molecules (purple), NAG (*N*-acetyl glucosamine) molecules (black)). Ligands are coloured according to the protomer to which they are bound (red or yellow). **e**–**h**, The orthosteric binding pocket of Ste2$_A$ (rainbow) showing ligands bound to Ste2–Ant (**e**), Ste2[IL]–Ag (**f**), Ste2[AL]–Ag (**g**) and Ste2–Ag–G (**h**). Densities were visualized using UCSF ChimeraX[36] and encompass a carve radius of 2 Å. The contour levels used for the maps were 0.0163 (**a**), 0.386 (**b**), 0.659 (**c**), 0.6 (**d**), 0.022 (**e**), 0.282 (**f**), 0.7 (**g**) and 0.03 (**h**). **h** is reproduced from ref. [1].

(superscripts denote class D1 numbering system[1]). Mutation of F204[5x28] and N205[5x29] in this region abrogates both ligand binding and signalling[1,5,6], highlighting the importance of the conformational change. Upon antagonist binding, the side chains of Phe204[5x28] flip inwards by 180° (7 Å inward shift of the Cα atom). Lys202[5x26] has to flip in the opposite direction to prevent it clashing with the ligand (Extended Data Fig. 5b), although there is only weak density for the side chain in Ste2–Ant because it extends into the detergent micelle (Extended Data Fig. 3g, h). Upon ligand binding, Phe204[5x28] makes extensive contacts with the antagonist and to α-factor in Ste2–Ag–G. Other side chains shift to prevent clashing with the ligand (for example, Tyr128[3x29], Asn132[5x29], Gln135[3x36], Asp274[ECL3] and Asp275[ECL3]) and Asn205[5x29] changes its orientation slightly to make better contacts with the antagonist (Extended Data Fig. 5b). When mutated, all of these residues have major effects on ligand binding and/or signalling[1,7–9]. The structure of Ste2–Ant also shares similarities with Ste2–Ag–G (r.m.s.d. 1.3 Å), but there are also some marked differences (Fig. 2a–d). The antagonist adopts the same hairpin shape as α-factor in the orthosteric binding pocket (Extended Data Fig. 5a), but makes contacts with fewer residues in the receptor (20 compared with 32 in Ste2–Ag–G; Extended Data Fig. 5c). The different sequences at the N terminus of the ligands define ligand efficacy[4] and they differ in structure substantially. The structure of the antagonist

N terminus bound to Ste2 is linear, with His2 forming a hydrogen bond with Asn205[5x29] on H5 and Ala3 interacting with Thr274[ECL3] in ECL3 (Fig. 2a). By contrast, the same region in α-factor is Y-shaped, which splays Trp1 and Trp3 wide apart (Fig. 2a), with Trp3 in a similar position to His2 in the antagonist and interacting with H5 (Lys202[5x26], Asn205[5x29] and Ile209[5x33]), whereas Trp1 interacts with H6 (Ile263[6x55] and Tyr266[6x58]). This is a key feature of receptor activation, because a 5.7 Å shift of the extracellular end of H6 (measured at Cα Ser267[6x59]) is required to accommodate this conformation of α-factor and is accompanied by a similar shift in the extracellular end of H7 (7.2 Å at Cα Gly-273[ECL3]) and a rearrangement of ECL3 (Fig. 2a). A consequence of the H6 movement upon agonist binding is a rotamer change of Tyr266, which displaces a water molecule and allows it to make interactions with α-factor and Asn205 that are not possible in the antagonist-bound state (Extended Data Fig. 8a).

To interrogate the early events of peptide binding that lead to receptor activation, we performed 3D variability analysis of the cryo-EM data (Extended Data Fig. 6a–d, Supplementary Videos 1–4). The 3D variability analysis algorithm[10] provides detailed visualization of the flexibility within a 3D reconstruction by analysing the conformational landscape of a protein molecule using a linear subspace model of 3D structures. Analyses of 3D data along the vectors of motion revealed key facets of peptide binding that were not evident in the consensus cryo-EM maps. In the Ste2–Ant state (Extended Data Fig. 6b), the receptor transitions from density maps that lack signal for the N terminus of the antagonist peptide to maps where this region becomes discernible, whereas the C terminus of the peptide is clearly visible. The stable engagement of the C-terminal region with the receptor is consistent with previous kinetic data suggesting that the C-terminus of the peptide engages with the receptor first[11]. By contrast, in the Ste2[IL]–Ag state (similar to the inactive Ste2–Ant state; the superscript denotes 'inactive-like' (IL)) (Extended Data Fig. 6c), densities for both the N-terminal and C terminal sections are visible in all maps along the principal components, supporting a stronger engagement of Trp1 to cause the shift of the extracellular region of H6 and the rotamer change of Tyr266[6x58]; by contrast, density in the middle of the peptide was weaker, presumably owing to its flexibility. In the 'active-like' (AL) agonist-bound Ste2[AL]–Ag state (similar to the active Ste2–Ag–G state; Extended Data Fig. 6d), data from all principal components indicated a stable engagement of the entire peptide within the orthosteric-binding pocket. This stable interaction of the peptide upon receptor activation is consistent with ligand-binding pocket (LBP) volume analysis (Extended Data Fig. 5e), which shows approximately 1,400 Å³ reduction from Ste2–LF to Ste2[AL]–Ag. This is similar to the trend observed in a class A GPCR[12], but is the

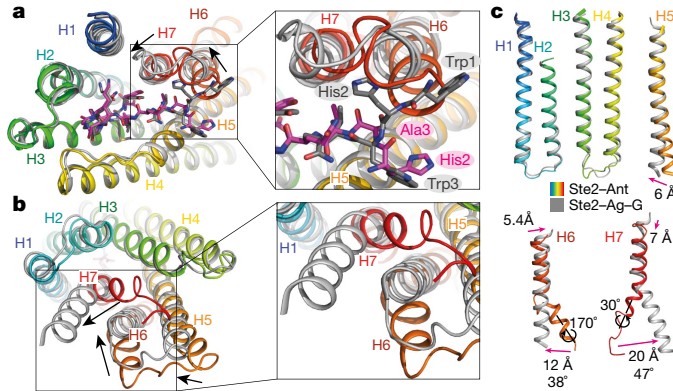

**Fig. 2 | Structural differences between Ste2–Ant and Ste2–Ag–G. a**, **b**, Superposition of Ste2–Ant (rainbow coloration) and Ste2–Ag–G (grey) with ligands shown as sticks (antagonist, purple carbon atoms; α-factor, grey carbon atoms). Major changes in secondary structure positions are indicated by arrows. **a**, Extracellular view. **b**, Intracellular view. **c**, Individual helices are shown after global alignment of Ste2 in different conformational states (colouration as in **a**).

opposite of the trend observed in a peptidergic class B GPCR, where a progressive expansion of approximately 1,200 Å[3] of the LBP occurs upon forming an active state[13].

The structural differences in the extracellular region between Ste2–Ant and Ste2–Ag–G are mirrored by substantial differences in the intracellular sections of H5, H6 and H7 (Fig. 2b). The intracellular end of H6 in Ste2–Ant is kinked outward away from the receptor core by 38° (12 Å shift at Cα Ser243[6x35]) compared with its position in Ste2–Ag–G, and this section is also rotated by 170° along the helix axis (Fig. 2c). In addition, the intracellular end of H5 is 6.6 Å further out from the receptor core than in the active state. H7 adopts a markedly different conformation in the two different states (Fig. 2b–d). In Ste2–Ag–G, the intracellular end of H7 is α-helical and kinks outwards from the receptor core by 20 Å (at Cα Asn301[7x61]) and makes contacts with the other protomer in the dimer. By contrast, in Ste2–Ant, H7 is kinked in towards the receptor core at an angle that differs by 47° from the active state and the residues Ala296[7x56]–Ser303[7x63] form a random coil. No density was observed for the 128 C-terminal residues in either structure, which are predicted to be unstructured[14] (TMHMM v2.0). The position of the intracellular end of H7 in Ste2–Ant overlaps with the position of the α5 helix[1] of the α-subunit Gpa1 in Ste2–Ag–G and thus sterically blocks G protein coupling. Notably, two putative sterol molecules reside at the kink in H7 in the inactive states of Ste2 (Extended Data Fig. 8b), but not in the active states; the sterol ergosterol is known to be important for Ste2 function[15].

## Agonist-bound states

In the cryo-EM images of ligand-free Ste2, Ste2–Ant and Ste2–Ag–G, there was only a single major conformation of the dimer. However, during processing of the Ste2–Ag dataset, two different conformations of Ste2 were found, and therefore two separate structures were determined (Extended Data Figs. 2, 3). One structure was very similar to Ste2–Ant (r.m.s.d. 0.7 Å over 263 Cα atoms, protomer A aligned) and is denoted Ste2[IL]–Ag. The other structure was very similar to Ste2–Ag–G (r.m.s.d. 0.45 Å over 290 Cα atoms, protomer A aligned) and is denoted Ste2[AL]–Ag (the superscript denotes 'active-like' (AL)). In addition, there were a number of classes similar to Ste2[AL]–Ag that were heterogeneous and could not be refined further (Extended Data Fig. 2; Methods). The active-like states (Ste2[AL]–Ag and heterogeneous Ste2[AL]–Ag) represented 84% of the particles analysed. Ste2–Ant and Ste2–Ag–G represent two extremes in the transition from an inactive state to an active state, and both Ste2[IL]–Ag and Ste2[AL]–Ag are consistent with being intermediate states (Fig 3a–c). The cryo-EM density for Ste2[IL]–Ag and Ste2[AL]–Ag shows lower resolution in H5–H7 than observed in Ste2–Ant and Ste2–Ag–G, resulting in only a proportion of the side chains being modelled (Extended Data Table 2), which highlights their dynamic role in the conformational change. On the extracellular face of the receptor, Ste2[IL]–Ag has a conformation part way between that of Ste2–Ant and Ste2–Ag–G. This conformation exhibits the agonist-induced displacement of H6 by 5.3 Å (measured at the Cα of Ser267[6x59]), although this is 0.4 Å short of its position in Ste2–Ag–G, and the reorientation of ECL3. However, there is no significant displacement of the extracellular end of H7 as observed in Ste2–Ag–G and neither are major changes on the intracellular face that are necessary for G protein coupling (Fig. 3a). These additional structural changes occur in the transition from Ste2[IL]–Ag to Ste2[AL]–Ag (Fig. 3b). The extracellular face of Ste2[AL]–Ag is almost identical to that of Ste2–Ag–G, with the full shift of H6 by 5.7 Å and H7 by 7.2 Å (at Cα Gly273[ECL3]). On the intracellular face, H7 in Ste2[AL]–Ag kinks outwards by 19.4 Å (at Cα Asn301[7x61]) and H6 moves inwards by 11.2 Å (at Cα Ser243[6x35]), which are only 0.6 Å and 0.7 Å short of their positions in Ste2–Ag–G. By contrast, there is a lateral movement in Ste2[AL]•Ag of the intracellular end of H5 (5.1 Å at Cα Arg234[5x58]).

Although Ste2[AL]–Ag has attained a conformation where the G protein coupling site is unblocked, there are additional structural changes

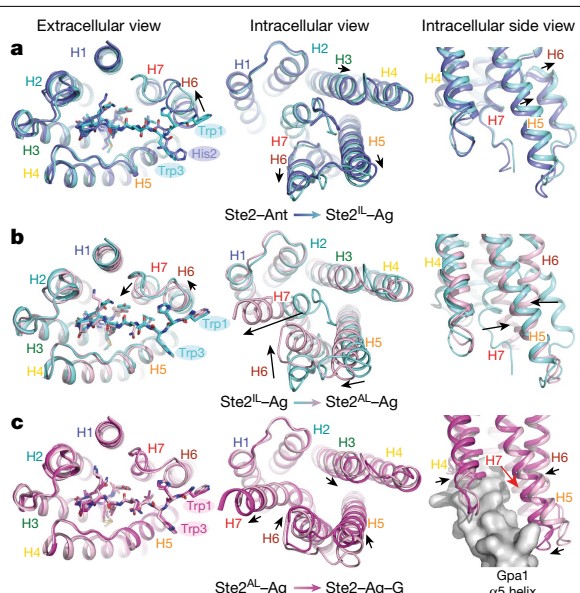

**Fig. 3 | Structural transitions upon activation of Ste2. a–c**, Superposition of Ste2–Ant (dark blue), Ste2[IL]–Ag (pale blue), Ste2[AL]–Ag (pink) and Ste2–Ag–G (magenta) during transition from Ste2–Ant to Ste2[IL]–Ag (**a**), Ste2[IL]–Ag to Ste2[AL]–Ag (**b**) and Ste2[AL]–Ag to Ste2–Ag–G.

required for full engagement of the G protein (Fig. 3c). The transition from Ste2[AL]–Ag to Ste2–Ag–G involves inward movements of H5 (4.7 Å at Cα Phe235[5x59]), H6 (1.2 Å at Cα Phe244[6x36]) and intracellular loop 3 (ICL3) and a lateral movement of H3, H4 and ICL2 by 1–2 Å. These movements result in repositioning of key residues, such as Asn158[ICL2] (1.8 Å at Cα), the side chain of Lys225[5x49] in H5, Leu247[6x39] in H6 (1 Å at Cα), and the side chain of Leu289[7x49] in H7, to enable favourable interactions with Gpa1. Mutations in all these residues strongly affect signalling[1].

Together, the five cryo-EM structures of Ste2 suggest the following activation mechanism. Agonist binding to the ligand-free inactive state of Ste2 causes reorientation of the extracellular end of H5 to facilitate ligand binding, an outward movement of the extracellular end of H6 and a shift of ECL3 (Ste2[IL]–Ag state), resulting in a contraction of the orthosteric binding site by an average of 27% (Extended Data Fig. 5e). This state is in equilibrium with the Ste2[AL]–Ag state, in which the extracellular end of H7 has moved inwards and the intracellular end of H7 has flipped outward from the G protein binding site, with the cytoplasmic end of H6 shifting inwards. These conformational changes form the α5-helix G protein binding site, further reduce the volume of the orthosteric binding site and increase the number of receptor–ligand contacts to H5 (Extended Data Fig. 5d). Engagement with a G protein stabilizes a further contraction of the α5 helix binding site through inward movements of the intracellular ends of H3, H4 and H5, resulting in a slight increase in volume of the orthosteric binding site and minor changes on the extracellular surface of the receptor.

## The dimer interface

Dimerization of Ste2 is essential for its activity[16,17]. During the activation process, the dimer interface remains intact but changes with respect to the residues making contacts and its surface area (Fig. 4a, b). The dimer interfaces are smaller in ligand-free Ste2, Ste2–Ant and Ste2[IL]–Ag (2,166 Å[2], 2,138 Å[2] and 2,145 Å[2], respectively) compared with Ste2[AL]–Ag and Ste2–Ag–G (2,727 Å[2] and 2,519 Å[2], respectively). Twenty-one residues in Ste2[A] form interfacial contacts in all the structures determined. Another seven residues form contacts only in Ste2[AL]–Ag and Ste2–Ag–G (Fig. 4a), owing to the interactions of the C-terminal end

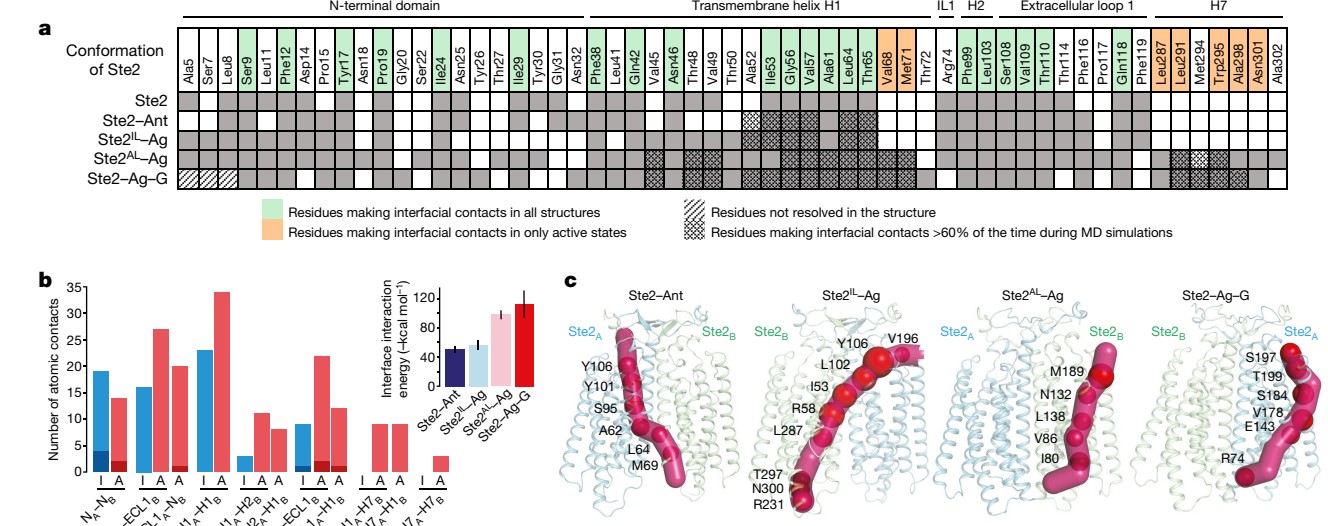

**Fig. 4 | Structural changes and allosteric communication across the dimer interface. a**, Residues at the dimer interface in the different conformational states (grey boxes), with residues making persistent contacts (more than 60% of molecular dynamics (MD) simulation time) indicated (cross-hatched boxes). IL1, intracellular loop 1. **b**, The total number of contacts between residues forming the dimer interface in the structures of the inactive-state (I, blue) Ste2–Ant and active-state (A, red) Ste2–Ag–G (van der Waals interactions (pale red and pale blue); hydrogen bonds (dark red and dark blue). Inset, the average (± s.d.) interface interaction energy (5,000 measurements from the last 200 ns of five independent simulations) for conformational states of the Ste2 dimer. **c**, Allosteric pipelines indicating the strongest correlated motions of amino acid residues. The top ten residues ranked by their contribution to allosteric communication within the dimers are shown as red spheres, with the radius reflecting their strength of contribution. The results were calculated from five independent 200-ns simulations.

of H7 with the cytoplasmic end of H1 in the adjacent protomer[1]. The increase in the number of contacts is also consistent with the increase in the calculated energy of interaction between the two protomers in the dimer (Fig. 4c). Molecular dynamics simulations of each of the individual states (see Methods) identified interfacial contacts that are present for more than 60% of the simulation time, and these also increased in number in the active states (Fig. 4a, Extended Data Figs. 7c, d). These changes in the dimer interface are accompanied by a relative shift of about a 3 Å of protomer B towards protomer A in the transition from the inactive to G protein-coupled state (Extended Data Fig. 8c). The dynamic nature of the dimer interface was not expected and suggested that there may be correlated residue movements that promote allosteric communication (cross-talk) between the two protomers. We therefore calculated statistical correlation in residue movements[18–20] for each of the dimer structures from the molecular dynamics simulation trajectories (Extended Data Fig. 7a, b) to identify the residue networks involved in allosteric communication. Of note, in Ste2–Ant and Ste2[IL]–Ag, the pipeline of residues showing the strongest correlations in their movement ran from the extracellular surface of one protomer, across the dimer interface, to the intracellular surface of H7 of the other protomer (Fig. 4d). By contrast, in the active states (Ste2[AL]–Ag and Ste2–Ag–G), the strongest correlated motions are within each protomer. Other weaker allosteric communication pipelines calculated for the four different states are shown in Extended Data Fig. 7b. G protein coupling affected allosteric communication within the Ste2 dimer, as seen by the differences between Ste2–Ag and Ste2–Ag–G, analogous to structural changes at the orthosteric binding site and extracellular surface observed upon coupling to a G protein[12]. Mutation of the residues that contribute to the allosteric communication can have an effect either on ligand binding (for example, Tyr101[2x63], Tyr106[D1e1x46], Ser184[4x64] and Tyr199[ECL2]) and/or receptor activation (for example, Ile53[1x43], Arg58[1x48], Met69[1x59], Ile80[2x42], Ser95[2x57], Leu102[2x64], Asn132[3x33], Glu143[3x44], Ser184[4x64], Thr199[ECL2] and Arg231[5x55]; see Supplementary Table 1 for details and references). Further work is necessary to identify the functional consequences of the allosteric communication across the dimer interface, which may also help the understanding of allosteric communication in class A and class B GPCR dimers.

## Role of conserved residues

Class A GPCRs contain the highly conserved NPXXY[7.53] motif on H7 which facilitates G protein binding through formation of a Tyr[5.58]–Tyr[7.53] hydrogen bond in the active state; this is coupled to outward movement of H6 (superscripts denote Ballesteros-Weinstein numbering[21]). Class D1 GPCRs contain the highly conserved LPLSSMWA motif (residues 289[7x49]–296[7x56]) on H7 in a similar position to the NPXXY[7.53] motif, but functions very differently (Extended Data Fig. 8d). Pro290[7x50] is conserved in 99% of Ste2 sequences and facilitates the formation of a $3_{10}$-helical region, which forms the kink in H7 in the inactive state. Mutations of residues in this motif have strong effects on the efficacy of signalling as it occupies the receptor core and functions as a central hub that mediates transition towards the active state by forming interactions with conserved residues in H3, H5 and H6 (Supplementary Table 1). The 47° kink at Pro290[7x50] in the inactive state of Ste2 is reminiscent of the 60° kink in H6 of class B GPCRs at the conserved sequence PxxG[6.50b] that is essential for the outward movement of its cytoplasmic end to form the G protein-binding cleft during receptor activation[22]. However, the sequence motifs around the respective Pro residues are different, as are the mechanisms of activation of class B and class D1 GPCRs.

Another highly conserved motif in class A GPCRs is the DRY motif, in which Arg[3.50] forms an ionic lock with H6 and stabilizes the inactive state. Ste2 does not share this motif, but Gln149[3x50] in H3 (99% conserved) occupies a position analogous to Arg[3.50] and appears to stabilize the inactive state of Ste2, but by a different mechanism (Extended Data Fig. 8e). In the inactive state, Gln149[3x50] makes a hydrogen bond with the conserved Ser292[7x52] of the LPLSSMWA motif (residues 289[7x49]–296[7x56]) in H7 and hydrophobic interactions with Met218[5x42], and Leu289[7x49] packs against Leu146[3x47]. Upon activation, Ser292[7x52] is replaced with Leu289[7x49], which then packs with Ile80[2x42] via van der Waals interactions, and this enables Leu289[7x49] to interact with Ile471[H5.25]

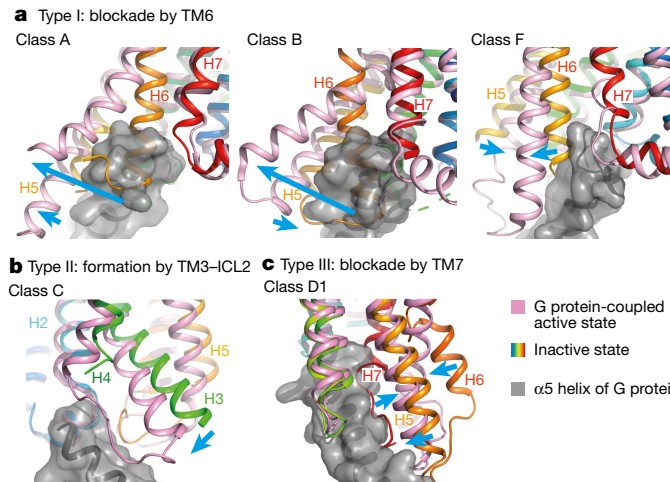

**a** Type I: blockade by TM6

Class A     Class B     Class F

**b** Type II: formation by TM3–ICL2

Class C

**c** Type III: blockade by TM7

Class D1

■ G protein-coupled active state
■ Inactive state
■ α5 helix of G protein

**Fig. 5 | Activation of different GPCR classes. a–c,** Representative examples of GPCRs from each of the major classes are shown in the inactive state (rainbow) and active state (magenta) coupled to a G protein (grey surface; only the C-terminal α5 helix is shown). Major changes in secondary structure upon receptor activation (blue arrows) form the binding site for the α5 helix of the G protein α-subunit through either the outward movement of helices from the binding site and/or movement of helices inwards to form the binding site. **a,** Class A, β₂-adrenoceptor[37,38] (Protein Data Bank (PDB) ID 2RH1 (active) and 3SN6 (inactive)); class B, corticotropin-releasing factor receptor 1[39,40] (PDB ID 4K5Y (active) and 6P9X (inactive)); and class F, smoothened receptor[33,41] (PDB IDs 5L7I (active) and 6OT0 (inactive)). **b,** Class C, type B GABA (γ-aminobutyric acid) receptor[34,42] (PDB IDs 7EB2 (active) and 7C7S (inactive)). **c,** Class D1, Ste2 receptor[1] (PDB IDs 7QA8 (active) and 7AD3 (inactive)).

(the superscript refers to the residue position in the common Gα numbering scheme for G proteins[23]) of the 'wavy hook' in the G protein α-subunit Gpa1 along with Gln149$^{3x50}$. Mutations in Gln149$^{3x50}$, Ser292$^{7x52}$, Leu289$^{7x49}$, Leu146$^{3x47}$ and Met218$^{5x42}$ cause strong constitutive activity, whereas Ile80$^{2x42}$ exhibits negative cooperativity[24–27] with Gln149$^{3x50}$ (Supplemental Table 1). Furthermore, side-chain changes in the conserved positively charged residues of H5 (K225$^{5x49}$ and R233$^{5x57}$) are necessary to allow G protein coupling and cause deficient signalling when mutated[28–30] (Extended Data Fig. 8h).

Pro258$^{6x50}$ is conserved in 98% of Ste2 sequences and facilitates the formation of a disordered region in H6, resulting in the C-terminal section of H6 pointing outwards in the inactive state so that it does not clash with the C-terminal section of H7 (Extended Data Fig. 8f). The kink in H6 packs against Leu291$^{7x51}$ and Ser292$^{7x52}$, which form a highly conserved L/I/F-S/G/T motif that is found in 95% of Ste2 sequences. The remainder of the C-terminal portion of H6 packs against H5. Pro258$^{6x50}$ is important for stabilizing the inactive conformation as suggested by the mutations P258L and P258C, which cause sevenfold and 47-fold increases in basal signalling, respectively[28,31] (Supplementary Table 1). Furthermore, the kink in H6 is coupled to a 180° flipping of the side chain of Gln253$^{6x45}$ that allows it to interact with Trp295$^{7x55}$ of the LPLSSMWA motif (Extended Data Fig. 8g). Upon activation, H6 assumes a straight α-helix that packs between H5 and H7, and Gln253$^{6x45}$ flips towards Ser288$^{7x48}$ as the conserved Trp295$^{7x55}$ in H7 moves away to enable H7 to make contacts with both H1 and H7 in the neighbouring protomer. Consistently, Gln253$^{6x45}$ and Pro258$^{6x50}$ are two of the strongest constitutively activating mutations in Ste2 and mutations in Ser288$^{7x48}$ of the LPLSSMWA motif lead to constitutive activity[8,27].

## Discussion

The activation mechanism of mammalian GPCRs (classes A, B, C and F) has been studied intensively, most recently through the structure

determination of many GPCR–G protein complexes by cryo-EM[32–34]. The paradigm for GPCR activation is that inactive states that cannot couple to a G protein undergo a conformational change facilitated by agonist binding that allows coupling to occur[2]. GPCRs appear to fall into three different categories (type I–III) depending on how the G protein binding site prevents coupling in the inactive state (Fig. 5). In class A and class B receptors (Fig. 5a), the primary block to G protein coupling is the cytoplasmic end of H6, which upon receptor activation moves away from the receptor core by 10–15 Å, forming a cleft in the cytoplasmic face of the receptor that binds the α5 helix of the G protein α-subunit[32]. In class F receptors (Fig. 5a), H6 blocks the G protein coupling site but requires a smaller movement (6 Å) to allow G protein coupling, because of the different angle of engagement of the α5 helix[33]. G protein coupling to class C receptors is very different (Fig. 5b), with the α5 helix engaging the periphery of the receptor primarily through ICL2, with the site being formed by a 5 Å movement of H3[34]. Class C receptors also provide the only previous data for how a GPCR dimer is activated, but dimerization and activation are mediated by the extracellular venus flytrap domains which are absent in all other GPCR classes. In addition, class C GPCR dimers are asymmetric and couple to a single G protein[35], in contrast to the symmetric dimer of Ste2, which can couple simultaneously to two G proteins[1]. Here we show that the class D1 receptor Ste2 has an activation mechanism distinct from other known GPCRs (Fig. 5c). Activation of Ste2 requires both the removal of a blockage from the G protein coupling site (H7) and formation of the binding site primarily through the inward movement of H6. In addition, Ste2 provides a model for how interactions at the dimer interface can change during receptor activation of transmembrane-mediated GPCR dimers, which could have implications for understanding signalling in other GPCR dimers.

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

# Methods

## Data reporting

Sample size was not predetermined by statistical methods, no randomization was applied to experiments and investigators were not blinded to experimental allocation or outcome assessment.

## Cloning and expression of wild-type Ste2

A construct encoding wild-type *S. cerevisiae* Ste2 (residues 1–431) that included a tobacco etch virus (TEV) cleavage site, eGFP, and a decahistidine tag (wtSte2-TEV-eGFP-His10) was synthesized as a gBlock gene fragment (IDT) and cloned into plasmid pAcGP67-B (BD Biosciences) by in vivo assembly in *Escherichia coli* XL10-Gold[43]. Expression of Ste2 in *Trichoplusia ni* High Five cells (Thermo Fisher Scientific; we did not test for mycoplasma) using high-titre recombinant baculoviruses prepared with the FlashBAC ULTRA system (Oxford Expression Technologies) was performed as described previously[1]. The cells were then collected by centrifugation, flash-frozen in liquid nitrogen, and stored at −80 °C until further use.

## Purification of Ste2 in the antagonist- and agonist-bound states

The antagonist α-factor peptide [des-Trp1, Ala3, Nle12]α-factor with the sequence HALQLKPGQP[Nle]Y was synthesized by Genscript. The biochemical antagonism of this peptide has been well-characterized[3,4,8,31]. Replacement of methionine in position 12 of α-factor with norleucine results in no change in biological properties[4]. The following procedure was used to purify antagonist-bound Ste2 and was adapted from our previously described procedure to purify agonist-bound Ste2[1]. All purification steps described below were carried out at 4 °C. Insect cell membranes from 2 litres of culture were prepared by three iterations of homogenization using an Ultra-Turrax disperser (IKA) in homogenization buffer (10 mM HEPES pH 7.5, 1 mM EDTA, 2 mM PMSF, 25 U ml$^{-1}$ benzonase, 10 mM MgCl$_2$, supplemented with EDTA-free protease inhibitor cocktail tablets (Roche), 1 mM benzamidine HCl, 2 µg ml$^{-1}$ aprotinin, 1 µg ml$^{-1}$ pepstatin, 0.5 mM AEBSF, 10 µg ml$^{-1}$ soybean trypsin inhibitor and 10 µM leupeptin) and ultracentrifugation at 125,000*g* for 90 min. The membranes were resuspended in solubilization buffer (20 mM HEPES pH 7.5, 100 mM NaCl, 20% glycerol, 2 mM PMSF, 25 U ml$^{-1}$ benzonase, 10 mM MgCl$_2$ and protease inhibitor cocktail) containing 10 µM antagonist α-factor peptide and incubated for 2 h. A final concentration of 1% (w/v) lauryl maltose neopentyl glycol (LMNG) (Anatrace), 0.1% (w/v) cholesteryl hemisuccinate (CHS) (Anatrace) was added to the membrane suspensions and incubated for 2 h. Ultracentrifugation (125,000*g* for 45 min) was performed to clarify the sample, and the solubilisate was supplemented with 8 mM imidazole, and mixed in batch with 10 ml Super Ni-NTA Agarose resin (Generon) for 2 h. The resin was packed by gravity-flow and washed with four column volumes of wash buffer (20 mM HEPES pH 7.5, 300 mM NaCl, 20% glycerol, 10 mM MgCl$_2$, 0.02% (w/v) LMNG, 0.01% (w/v) CHS, 1 µM antagonist α-factor, 40 mM imidazole) and was eluted with elution buffer (20 mM HEPES pH 7.5, 100 mM NaCl, 20% glycerol, 10 mM MgCl$_2$, 0.02% (w/v) LMNG, 0.01% (w/v) CHS, 1 µM antagonist α-factor, 300 mM imidazole). The eluate was incubated overnight with 2.5 mg TEV protease and 1 mM DTT. The cleaved sample was exchanged into desalting buffer (20 mM HEPES pH 7.5, 100 mM NaCl, 20% glycerol, 10 mM MgCl$_2$, 0.02% (w/v) LMNG, 0.01% (w/v) CHS, 1 µM antagonist α-factor, 5 mM imidazole) using Sephadex G-25 PD-10 desalting columns (GE Healthcare). Then the TEV protease and the cleaved eGFP-His10 were removed by negative purification on TALON resin (Takara Bio). The flow-through fraction was concentrated using a 50 kDa molecular weight cut-off Amicon Ultra centrifugal concentrator (Merck) and loaded at 0.35 ml min$^{-1}$ onto an Agilent Bio SEC-5 500 Å column, pre-equilibrated in 20 mM HEPES pH 7.5, 100 mM NaCl, 2 mM MgCl$_2$, 0.01% (w/v) GDN and 5 µM antagonist α-factor. The peak fractions containing Ste2 were pooled and concentrated. Protein concentration was estimated by NanoDrop 2000 (Thermo Scientific) at 280 nm using an extinction coefficient of 1.157 ml mg$^{-1}$ cm$^{-1}$.

Purification of Ste2 in the agonist-bound state was performed as described above except the antagonist α-factor peptide was replaced with agonist α-factor peptide with the sequence WHWLQLKPGQPMY (Genscript).

## Purification of Ste2 in the ligand-free state

To purify Ste2 in the ligand-free state, we devised PSGWAY, which involves addition of purified wild-type Gpa1–Ste4–Ste18 heterotrimer to stabilize ligand-free Ste2 before solubilization from cell membranes. Since the agonist α-factor was not present, the wild-type Gpa1–Ste4–Ste18 heterotrimer transiently stabilized Ste2 during detergent solubilization, but did not form long-term stable interactions with the receptor, thereby allowing purification of ligand-free Ste2 in reasonable yields that were not achieved in the absence of wild-type G protein heterotrimer. A similar approach has recently been used to purify the class B1 CGRP receptor in the ligand-free state for cryo-EM structure determination, however in that study a specific mutant Gα$_{11}$ protein was co-expressed in insect cells to transiently stabilize the ligand-free CGRP receptor[13]. Purification of wild-type Gpa1–Ste4–Ste18 heterotrimer was performed as described previously[1]. In brief, wild-type Gpa1 was expressed and purified from *E. coli*. Ste4–Ste18 was co-expressed and purified from *T. ni* cells (Expression Systems) using the flashBAC ULTRA system (Oxford Expression Technologies). The wild-type Gpa1–Ste4–Ste18 heterotrimer complex was formed by mixing Gpa1 and Ste4–Ste18 dimer at 1:1 molar ratio and the complex was loaded and separated on Superdex 200 10/300 GL column (GE Healthcare) pre-equilibrated with 10 mM HEPES pH 7.5, 100 mM NaCl, 10% glycerol, 1 mM MgCl$_2$ and 0.01 mM GDP. Peak fractions were pooled and the complex was concentrated in a 10 K molecular weight cut-off Amicon ultra centrifugal concentrator (Merck). For stabilizing ligand-free Ste2, insect cell membranes with Ste2 were incubated with 5.7 mg of purified wild-type Gpa1–Ste4–Ste18 heterotrimer supplemented with 400 mU ml$^{-1}$ apyrase for 2 h. Solubilization and purification of ligand-free Ste2 was then performed as described above for the antagonist-bound Ste2, except no α-factor peptide was added throughout purification. As wild-type Gpa1-Ste4-Ste18 heterotrimer only transiently interacts with Ste2 in the absence of α-factor, the Ste2 eluate from Ni-NTA column did not contain any G protein heterotrimer as it had been eliminated in the washing steps. Only limited structural information about ligand-free states of GPCRs is currently available[13] and the PSGWAY method described here could possibly be adapted to purify ligand-free states of other GPCRs.

## Vitrified sample preparation and data collection

Cryo-EM grids were prepared by applying 3 µl of purified agonist-bound Ste2 (4.7 mg ml$^{-1}$), antagonist-bound Ste2 (3.1 mg ml$^{-1}$) and ligand-free Ste2 (3 mg ml$^{-1}$) onto glow-discharged holey gold (Quantifoil Au 1.2/1.3 300) mesh. The grids were blotted with filter paper for 2.5 s before plunge-freezing in liquid ethane (at −181 °C) using an FEI Vitrobot Mark IV at 100% relative humidity and 4 °C. All cryo-EM datasets were collected on an FEI Titan Krios microscopes operating at 300 kV. For the antagonist-bound Ste2 dataset, images were recorded on a K3 direct electron detector in counting mode post quantum energy filter (Gatan) operated in zero-energy-loss mode with a slit width of 20 eV to obtain dose-fractionated movies of the sample with a 100 µm objective aperture. 15,751 micrographs were recorded at a magnification of 105,000× (0.86 Å per pixel; LMB Krios3) as dose-fractionated movie frames with an exposure time of 2.08 s amounting to a total exposure of 57 e$^-$ Å$^{-2}$ and defocus range set between −0.7 and −2.0 µm. For the ligand-free Ste2 dataset, images were recorded on a Falcon 4 direct electron detector (Thermo Fischer Scientific) operated in Electron Event Representation (EER) mode with a 100 µm objective aperture. 9,369 micrographs were recorded at a magnification of 96,000× (0.85 Å per pixel; LMB Krios2) in super-resolution mode with an exposure time of 8.81 s amounting to

to a total exposure of 54 e⁻ Å⁻² and defocus range set between −0.8 and −2.4 μm. For the agonist-bound Ste2 dataset, images were recorded on a K2 direct electron detector in counting mode post quantum energy filter (Gatan) operated in zero-energy-loss mode with a slit width of 20 eV to obtain dose-fractionated movies of the sample with a 100 μm objective aperture. 6,944 micrographs were recorded at a magnification of 105,000× (1.1 Å/pix; LMB Krios2) as dose-fractionated movie frames with an exposure time of 12.5 s amounting to a total exposure of 50 e⁻ Å⁻² and defocus range set between −0.9 and −2.7 μm.

## Cryo-EM data processing and 3D reconstruction

Image stacks (15,751 antagonist-bound, 9,369 ligand-free and 6,944 agonist-bound Ste2 movies) were subjected to beam-induced motion correction using MotionCor2[44] by dividing each frame into 5 × 5 patches. CTF parameters were estimated from non-dose-weighted micrographs in GCTF[45] with equiphase averaging for antagonist-bound and agonist-bound Ste2 datasets and CTFFIND-4.1 in RELION3.1[46] for ligand-free Ste2 dataset. Autopicking was performed using BoxNet deep convolutional neural network implemented in Warp[47] that yielded 4,982,771 particles, 2,409,127 particles and 1,854,854 particles for the antagonist-bound, ligand-free and agonist-bound Ste2 datasets, respectively. Particles were extracted in a box-size equivalent to 190 Å and down-scaled initially to 3.3 Å per pixel. An ab initio 3D model was generated using stochastic gradient descent algorithm implemented in RELION3.1. The extracted particles were subjected to two rounds of 3D classification in C1 symmetry in RELION3.1 and the particles that displayed clear transmembrane features were selected which yielded 1,285,901 particles, 463,024 particles, and 399, 652 particles for the antagonist-bound, ligand-free and agonist-bound Ste2 datasets, respectively. These particles were consistent with a homodimeric Ste2 in all three datasets and no reliable classes consistent with a Ste2 monomer were obtained despite extensive 2D and 3D classifications. These particles were re-extracted in a box-size equivalent to 210 Å and subjected to 3D reconstruction in C1 symmetry followed by iterative rounds of Bayesian polishing, beam-tilt correction and per-particle CTF refinement in RELION-3.1. Subtraction of the detergent micelle signal was then performed and the particles were subjected to 3D classification without alignment. For the antagonist-bound Ste2 dataset, 136,877 particles that displayed high-resolution features were selected and subjected to Bayesian polishing, per-particle CTF refinement using the improved model and 3D reconstruction in C2 symmetry which yielded a map that refined to a global resolution of 2.69 Å (Fourier shell correlation (FSC) = 0.143). For the ligand-free Ste2 dataset, 67,791 particles that refined to a high-resolution were selected and the detergent micelle signal was reverted. The particle stack was subjected to Bayesian polishing and per-particle CTF refinement using the improved model. The particle stack was then exported to cryoSPARC v3.1[48] where they underwent non-uniform 3D refinement[49] in C2 symmetry which yielded a map that refined to a global resolution of 3.10 Å (FSC = 0.143). For the agonist-bound Ste2 dataset, two distinct classes were obtained from 3D classification without alignment: one class with 64,091 particles that resembled the antagonist-bound and ligand-free Ste2 in the inactive state (therefore called Ste2^IL) and another class with 65,284 particles that resembled the G-protein heterotrimer coupled Ste2 structure in the active state (therefore called Ste2^AL). The three other classes obtained from 3D classification without alignment were heterogenous and it was not possible to separate into distinct intermediate states upon further 3D classification. However, these heterogenous classes were more similar to the active-like state since TM7 showed relatively strong density facing the dimer interface, which is observed only in the active-like and G protein heterotrimer coupled active states of Ste2. Thus, in the presence of the agonist α-factor, the majority of particles (83.9%) correspond to an active-like state or intermediate states transitioning towards the active-like state and only a minority (16.1%) of the particles correspond to a distinct inactive-like state which is consistent with α-factor binding transitioning Ste2 into an active-like state that is conducive to G protein coupling. The distinct set of 64,091 particles (Ste2^IL state) and 65,284 particles (Ste2^AL) were refined separately and showed high-resolution features corresponding to these distinct intermediate states. The detergent micelle signal was reverted and these separate particle stacks were subjected to Bayesian polishing and per-particle CTF refinement. The particles were then exported to cryoSPARC v3.1 where they underwent non-uniform 3D refinement in C2 symmetry that yielded maps at global resolutions of 3.53 Å and 3.46 Å for the inactive-like and active-like agonist-bound Ste2, respectively (FSC = 0.143). C2 symmetry was applied to the final reconstructions, after 3D classifications, since no significant differences between the two protomers were observed in the high-resolution cryo-EM maps obtained in C1 symmetry. In addition, application of C2 symmetry improved the resolution of the maps by 0.2–0.3 Å, which is expected to occur only if the two protomers are identical. The improved map resolution increased confidence in model building in regions with relatively lower local resolution in the intermediate state Ste2^IL–Ag and Ste2^AL–Ag structures. The refined particle stacks that contributed to each of the four final maps described above were used for 3D-Variability analysis[10] as implemented in cryoSPARC v3.1. A generous mask generated with a 5 pixel map expansion and 10 pixel soft edge was used to capture any possible motions during the 3D variability analysis. For each particle stack corresponding to one of the four distinct states of Ste2 (antagonist-bound, ligand-free, agonist-bound inactive-like, agonist-bound active-like Ste2), four orthogonal principle modes (eigenvectors of the 3D covariance) were solved. The resulting volume frame data generated in CryoSPARC v3.1 were examined in UCSF Chimera[50] as volume series and captured as movies. Local resolution was determined in cryoSPARC v3.1 that uses a local windowed FSC method similar to the blocres program of Bsoft package[51]. Postprocessed maps were generated in either RELION or cryoSPARC v 3.1. Postprocessed maps shown in Extended Data Fig. 3a–d was generated using the Deep-EMhancer algorithm[52].

## Structure determination and model refinement

The model of Ste2 receptor portion of Ste2–miniGpa1–Ste4–Ste18 complex (PDB: 7AD3) was used as an initial template and was fitted into the cryo-EM density maps in UCSF ChimeraX[36]. Portions of the receptor that differed from the initial model were rebuilt manually in COOT[53] followed by iterative rounds of refinements in CCP-EM[54] and PHENIX[55] software suites and manual model building in COOT. Restraints for NAG and CHS (monomer library ID Y01) were derived in eLBOW[56] and AceDRG[57], respectively. C2 symmetry constraints were applied during model refinements. Comprehensive model validation was performed in PHENIX and MolProbity[58]. The final models achieved good geometry (Extended Data Table 1). No density was visible for the N-terminal residues 1–4 and C-terminal residues 304–431. For the Ste2–Ant, Ste2^AL–Ag and Ste2^IL–Ag states, density modification was performed on unfiltered half-maps using phenix.resolve_cryo_em[59] for better visualization of some weaker map regions, and where present water molecules, to assist model building in COOT; however all refinements and map validations were performed against the original EM map.

## Figures

All figures were generated using either PyMOL[60] (v2.5), UCSF Chimera[50] (v1.15) and UCSF ChimeraX[36]. Protein backbone in 3D variance analysis figures (Extended Data Fig. 6) was flexibly fitted into map frames in COOT. Binding pocket volumes (Extended Data Fig. 5e) were calculated using HOLLOW v1.3[61] and visualized using UCSF Chimera.

## Molecular dynamics simulations

All-atom molecular dynamics simulations were performed using the CHARMM36m forcefield[62] and GROMACS MD package. The simulations were started from the cryo-EM structures of the Ste2 dimer in Ste2–Ant,

Ste2[II]–Ag, Ste2[AL]–Ag and Ste2–Ag–G. The structures of Ste2–Ant. Ste2[II]–Ag and Ste2[AL]–Ag were prepared using Maestro protein-preparation wizard (Schrödinger Release 2021-2), including adding missing heavy atoms and all hydrogens, followed by full energy minimization. Histidine residues in the ligands were in the uncharged imidazole form. The CHS molecules from the cryo-EM structures were aligned and replaced by cholesterol using Pymol. The protein-cholesterol complexes were taken as initial input for CHARMM-GUI lipid bilayer builder[63–65] and placed in a $9 \times 9$ nm$^2$ POPC bilayer. The resulting protein-lipid complexes were solvated with water molecules to a thickness of 1.4 nm from the receptor along the z-axis. The charges in the systems prepared were neutralized with 0.15 M concentration of NaCl. The neutralized systems were first heated from 0 K to 310 K in NVT ensemble using a Nosé-Hoover thermostat[66] in 0.2 ns. A restraint force of 5 kcal mol$^{-1}$Å$^{-2}$ was applied on the heavy atoms of protein and lipid during the heating process. The restraints force was gradually reduced to 0 kcal mol$^{-1}$Å$^{-2}$ in a 30 ns equilibration protocol performed with an NPT ensemble, with a 1 kcal mol$^{-1}$Å$^{-2}$ per 5 ns window. The pressure was controlled using Parrinello-Rahman method[67] and the simulation systems were coupled to a one bar pressure bath. The last frames from the equilibrium protocol were taken as the initial structure for the regular molecular dynamics production run. Five random seeds were used to assign initial velocities to start five production molecular dynamics simulation runs for each Ste2 dimer system. The production run was 400 ns long for each velocity. In all simulations, the LINCS algorithm was applied on all bonds and angles of water. The integration time step was set to 2 fs. A cut-off of 1.2 nm was used for non-bond interactions and the particle mesh Ewald method (PME)[68] was applied to treat long-range Lennard-Jones interactions. The MD trajectories were stored at every 20 ps interval. The 200 to 400 ns production trajectories of each velocity (200 ns × 5 = 1,000 ns total) were combined for each conformation of Ste2 dimer for analysis.

The inter transmembrane helix residue contact analysis was carried out using the script get_contact (https://getcontacts.github.io/), that calculates different types of residue contacts including hydrogen bond, van der Waals, salt bridge and cation–pi contacts. The definition of the range of residues in each transmembrane helix was taken from previous work[1]. Each pair of the transmembrane helices was input as two selection groups to the get_contact script to calculate the contact formed during dynamics and the frequency of each of these contacts across the whole molecular dynamics trajectories. A pair of residues is considered to form a sustained contact during simulation if the contact frequency is greater than 60%.

The interaction energies between two monomers for each system were calculated as averaged over the molecular dynamics simulation trajectories with the GROMACS energy module. The interaction energy calculation was calculated between residues in TM1 and TM7 from one monomer with the residues in TM1 and TM7 of the other monomer. The interaction energy was calculated as the sum of short-range van der Waals interaction energy and Columbic interaction energy.

Allosteric communication residue network analysis used the in-house software[20,69] Allosteer to identify the network of residues involved in allosteric communication from the extracellular ligand binding region to the intracellular G protein coupling residue site in the dimer. We calculated the mutual information in torsional angle distribution for all pairs of residues in the Ste2 dimers. Then we used graphic theory to calculate the shortest pathway with highest mutual information connecting distant pairs of residues with high mutual information. We calculated the allosteric communication pathways from extracellular residues, going through the residues in the ligand binding site to the G protein coupling site residues. Previously we have shown that mutation of the residues predicted to be in the allosteric communication pipelines have shown change in receptor coupling to G proteins and/or β-arrestin in class A GPCRs[19,69,70]. The list of residues in the Ste2 ligand binding site used for calculating the allosteric communication pathways were chosen based on their contact frequency with the ligand during dynamics. All the residues that showed greater than 60% frequency in contacts with the ligand were taken as ligand-binding site residues. Similarly, G protein binding site residues were also defined by contact analysis on a Ste2–Ag–G simulation. The pipelines between these two groups were sorted by their strength, and their rank reflects the strength of allosteric communication. The number of allosteric communication pathways passing through each residue is defined as a 'hubscore' and this is used to describe the contribution by each residue to the strength of the allosteric communication.

### Reporting summary

Further information on research design is available in the Nature Research Reporting Summary linked to this paper.

### Data availability

Structures have been deposited in the Protein Data Bank (PDB; https://www.rcsb.org/), and the associated cryo-EM data have been deposited in the Electron Microscopy Data Bank (EMDB; https://www.ebi.ac.uk/pdbe/emdb/) and the Electron Microscopy Public Image Archive (EMPIAR; https://www.ebi.ac.uk/empiar/): ligand-free Ste2 (7QB9, EMD-13882 and EMPIAR-10878); Ste2–Ant (7QA8, EMD-13880 and EMPIAR-10877); Ste2[II]–Ag (7QBC, EMD-13886 and EMPIAR-10879); Ste2[AL]–Ag (7QBI, EMD-13887 and EMPIAR-10879). There are no restrictions on data availability.

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

**Acknowledgements** V.V. was funded by a Gates Cambridge Scholarship. The work in the laboratory of C.G.T. was supported by core funding from the Medical Research Council, as part of United Kingdom Research and Innovation (MRC U105197215). The work in the laboratory of N.V. was funded by grants from the National Institutes of Health (2R01-GM117923). We thank Y. Lee, T. Nakane, R. Henderson, K. Naydenova, C. J. Russo, P. Edwards and T. Warne for helpful discussions; and the LMB electron microscopy facility and J. Grimmett and T. Darling from LMB scientific computing for technical support during this work.

**Author contributions** V.V. conceptualized and developed the PSGWAY purification method, performed receptor and G protein expression, purification, preparation of cryo-EM grids, cryo-EM data collection, data processing, structure determination and model building. N.M. performed the molecular dynamics simulations, and N.M. and N.V. did the analysis of the molecular dynamics simulation trajectories. V.V. and C.G.T. carried out structure analysis and manuscript preparation. C.G.T. managed the overall project. The manuscript was written by V.V. and C.G.T., and included contributions from all the authors.

**Competing interests** The authors declare no competing interests.

**Additional information**
**Correspondence and requests for materials** should be addressed to Christopher G. Tate.

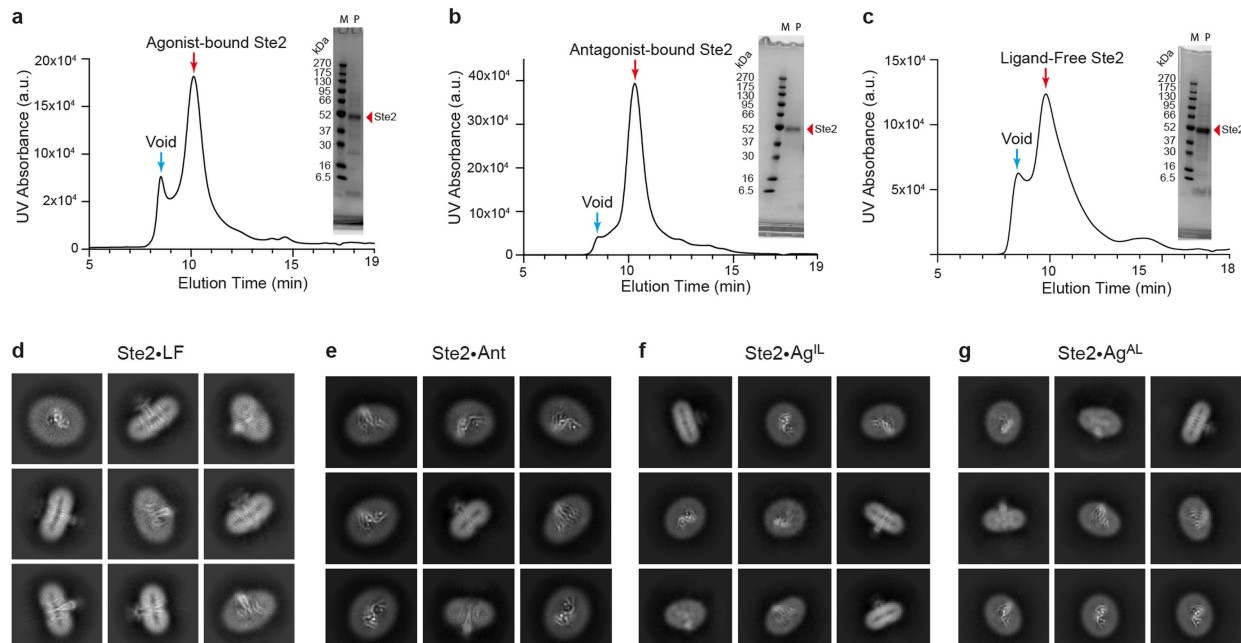

**Extended Data Fig. 1 | Purification and structure determination of antagonist-bound, ligand-free and agonist-bound Ste2. a-c**, Representative size-exclusion chromatography traces and SDS-PAGE analysis of purified agonist-bound, antagonist-bound and ligand-free Ste2, respectively. The peak fraction (red arrows) was used for SDS-PAGE analysis. Material eluting in the void volume comprised of aggregates (blue arrow). On the SDS-PAGE gels, lanes contain molecular weight markers (M) and purified Ste2 (P). Experiments were repeated three times for agonist-bound Ste2, twice for antagonist-bound and twice for ligand-free Ste2 with similar results. **d–g**, 2D class averages of the cryo-EM data for antagonist-bound, ligand-free, agonist-bound inactive-like and agonist-bound active-like Ste2, respectively.

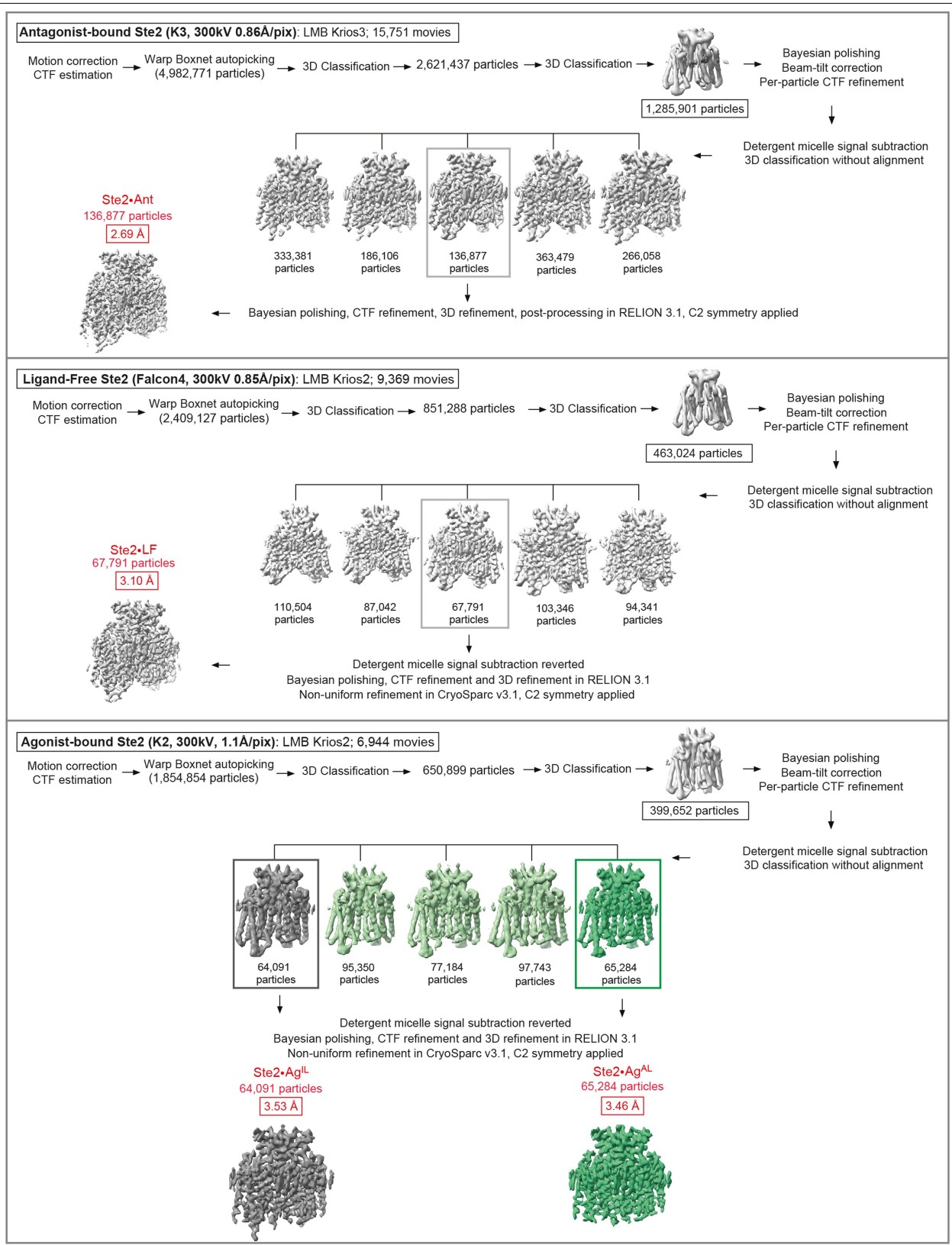

**Extended Data Fig. 2** | See next page for caption.

**Extended Data Fig. 2 | Flow chart of single-particle cryo-EM data processing.** Micrographs (15,751 for antagonist-bound, 9,369 for ligand-free, 6,944 for agonist-bound Ste2) were collected from independent sessions on a Titan Krios operated at 300 kV. Each dataset was corrected for drift, beam-induced motion and radiation damage. After estimation of CTF parameters, particles were autopicked and subjected to two rounds of 3D classification. The number of good particles selected from each round of 3D classification is shown. The initial set of best particles was subjected to Bayesian Polishing, beam-tilt correction and per-particle CTF refinement. After detergent signal subtraction, 3D classification without alignment was performed. The resulting best class of particles was selected and subjected to Bayesian Polishing and per-particle CTF refinement. For ligand-free Ste2 and agonist-bound Ste2 datasets, non-uniform 3D refinement was performed after reverting detergent signal. For agonist-bound Ste2, a distinct inactive-like (grey) and active-like (sea green) state cryo-EM maps were obtained. Three additional classes that were heterogenous (pale green) and tending towards the active-like state were obtained (see methods). The global resolution of the final maps was calculated using gold-standard FSC of 0.143.

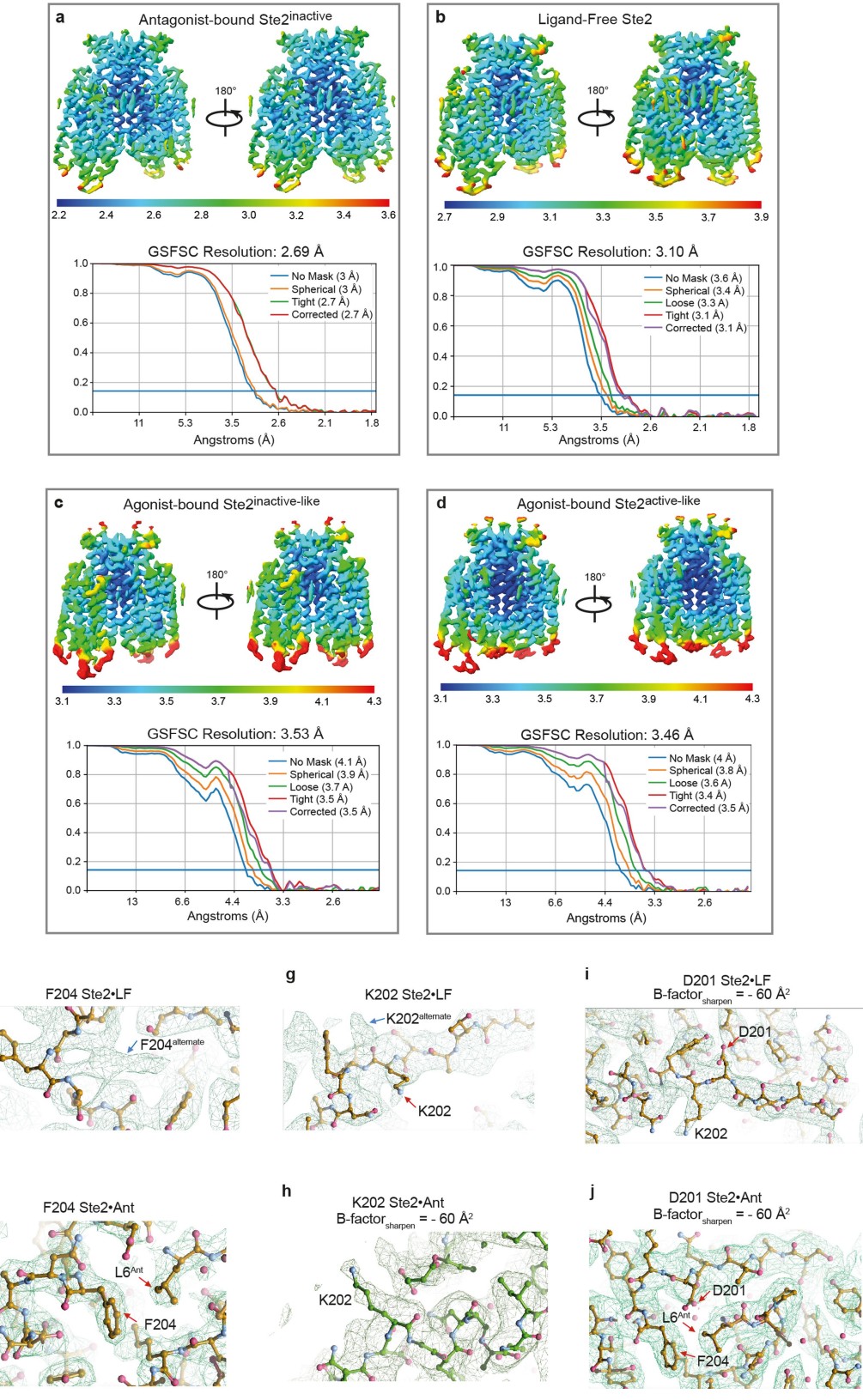

**Extended Data Fig. 3** | See next page for caption.

**Extended Data Fig. 3 | Local resolution and gold-standard Fourier shell correlation curves of cryo-EM maps. a-d**, Cryo-EM maps coloured according to local resolution from highest resolution (dark blue) to lowest resolution (red) are shown along with gold-standard Fourier shell correlation (GSFSC) curves for the final maps and map validation from half maps. The GSFSC curves show overall nominal resolutions of 2.69 Å for Ste2•Ant, 3.10 Å for Ste2•LF, 3.53 Å for Ste2•Ag$^{IL}$, and 3.46 Å for Ste2•Ag$^{AL}$. **e**, Two densities potentially attributable to F204 was visible in the Ste2•LF state and the side chain was assigned to the stronger density (contour level 0.24). **f**, Only one clear density attributable to F204 was visible in the Ste2•Ant state and this side chain interacts with the antagonist α-factor (contour level 0.036). **g**, Two densities potentially attributable to K202 was visible in the Ste2•LF state and the side chain was assigned to the stronger density (contour level 0.17). **h**, Only one density that protrudes into the detergent micelle is visible for K202 in the Ste2•Ant state (contour level 0.01). **i, j**, A weak density projecting outwards from the ligand binding pocket is visible for D201 in the Ste2•LF state (contour level 0.2), whereas a density attributable to D201 projects towards and makes interactions with the antagonist α-factor in the Ste2•Ant state (contour level 0.01). Since the density for D201 is weak in the Ste2•LF state, we have stubbed this residue (Extended Data Table 2).

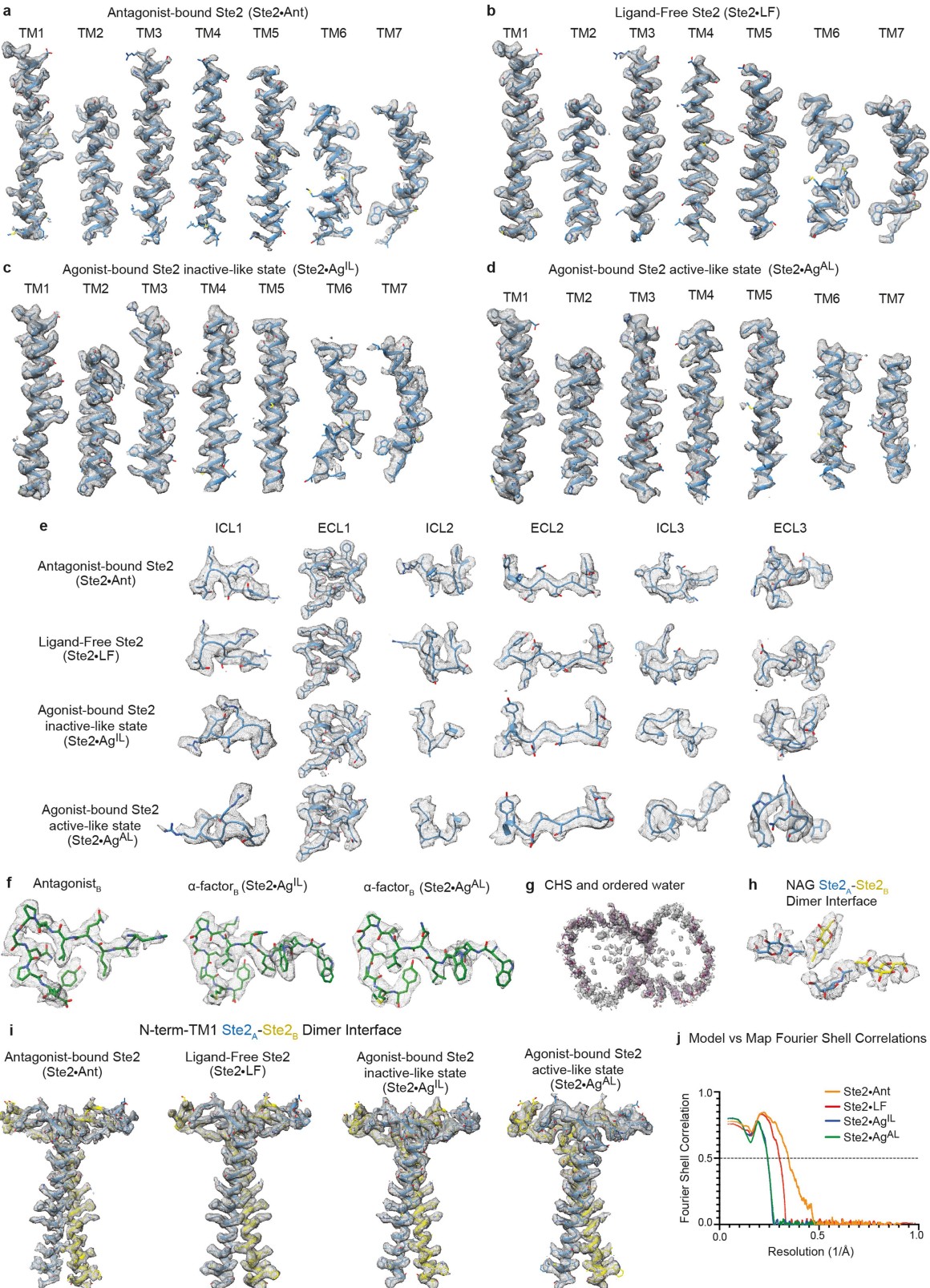

**Extended Data Fig. 4 |** See next page for caption.

**Extended Data Fig. 4 | Atomic modelling of Ste2 structures in the cryo-EM density maps. a-d**, Densities were visualized using UCSF ChimeraX[36] and encompass a carve radius of 2 Å around the indicated region. Density maps and models are shown for all seven transmembrane helices of Ste2•Ant (contour level 0.024), Ste2•LF (contour level 0.62), Ste2•Ag$^{IL}$ (contour level 0.6 for TM1-TM6 and 0.49 for TM7), and Ste2•Ag$^{AL}$ (contour level 0.6 for TM1-TM6 and 0.49 for TM7), respectively. **e**, Densities for intracellular and extracellular loops (ICL and ECL) 1, 2 and 3 are shown for the indicated structures. Ste2•Ant is contoured at 0.0125 (except ECL1 contoured at 0.02). Ste2•LF is contoured at 0.3 (except ICL1 contoured at 0.45 and ECL1 contoured at 0.6). Ste2•Ag$^{IL}$ is contoured at 0.4 (except ECL1 and ECL2 contoured at 0.6). Ste2•Ag$^{AL}$ is contoured at 0.6 (except ICL2 and ICL3 contoured at 0.4). **f**, Densities for antagonist$_B$ (contour level 0.022), α-factor$_B$ in Ste2•Ag$^{IL}$ (contour level 0.6) and α-factor$_B$ in Ste2•Ag$^{AL}$ (contour level 0.9) are shown as a mesh. **g**, A belt of putative CHS molecules surrounding Ste2•Ant are shown along with modelled ordered water molecules (contour level 0.018). **h**, Densities for N-acetylglucosamine molecules attached to Asn25 and Asn32 of Ste2$_A$ and Ste2$_B$ were present in all cryo-EM maps, and representative densities from Ste2•Ant are shown (contour level 0.015). Asn25 and Asn32 are well-characterized N-glycosylation sites in Ste2[1,71]. **i**, Densities for N-terminal domain and TM1 dimer interface are shown for Ste2•Ant (contour level 0.04), Ste2•LF (contour level (contour level 0.62), Ste2•Ag$^{IL}$ (contour level 0.7), and Ste2•Ag$^{AL}$ (contour level 0.7). **j**, Model versus Map Fourier shell correlation (FSC) curves obtained from validation of atomic models against cryo-EM maps are shown. At FSC = 0.5, model resolutions were 2.9 Å, 3.3 Å, 3.8 Å, and 3.7 Å for Ste2•Ant, Ste2•LF, Ste2•Ag$^{IL}$ and Ste2•Ag$^{AL}$, respectively.

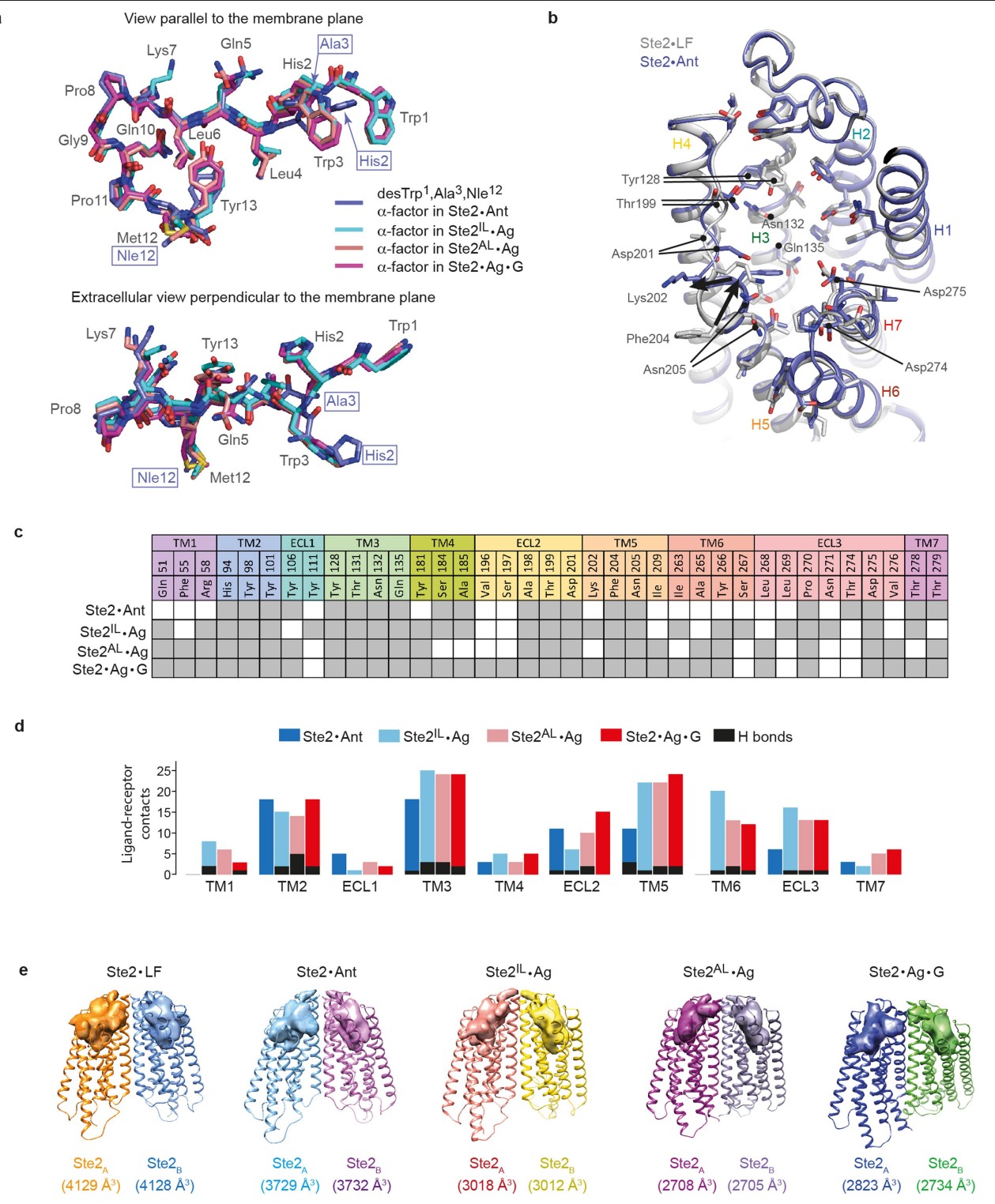

**Extended Data Fig. 5 | Ligand structures and the orthosteric binding pocket. a**, Superposition of ligand structures in each of the Ste2 structures determined. Amino acid residues in α-factor are all labelled (black), whereas only residues in the antagonist that differ from α-factor are labelled (blue boxes). **b**, Superposition of Ste2•LF (grey) and Ste2•Ant (purple) to highlight changes in the position of residues (sticks) within the orthosteric binding site. **c**, Amino acid residues that interact with the ligand (grey boxes) in Ste2 structures. **d**, Number of receptor-ligand contacts determined for each structural element in Ste2 and how they change during receptor activation. **e**, Changes in the orthosteric binding site volume (surface representation) during Ste2 (cartoon representation) activation. Analyses were performed in the absence of the ligand and extracellular N-terminal domain coordinates (residues 1–37). Binding pocket volumes were calculated using the software HOLLOW[61] and the results were visualized in Chimera. The PDB output (cavity filled dummy atoms) from HOLLOW were converted to a map and the volumes of the binding pocket were measured using Chimera's "Measure Blob" function.

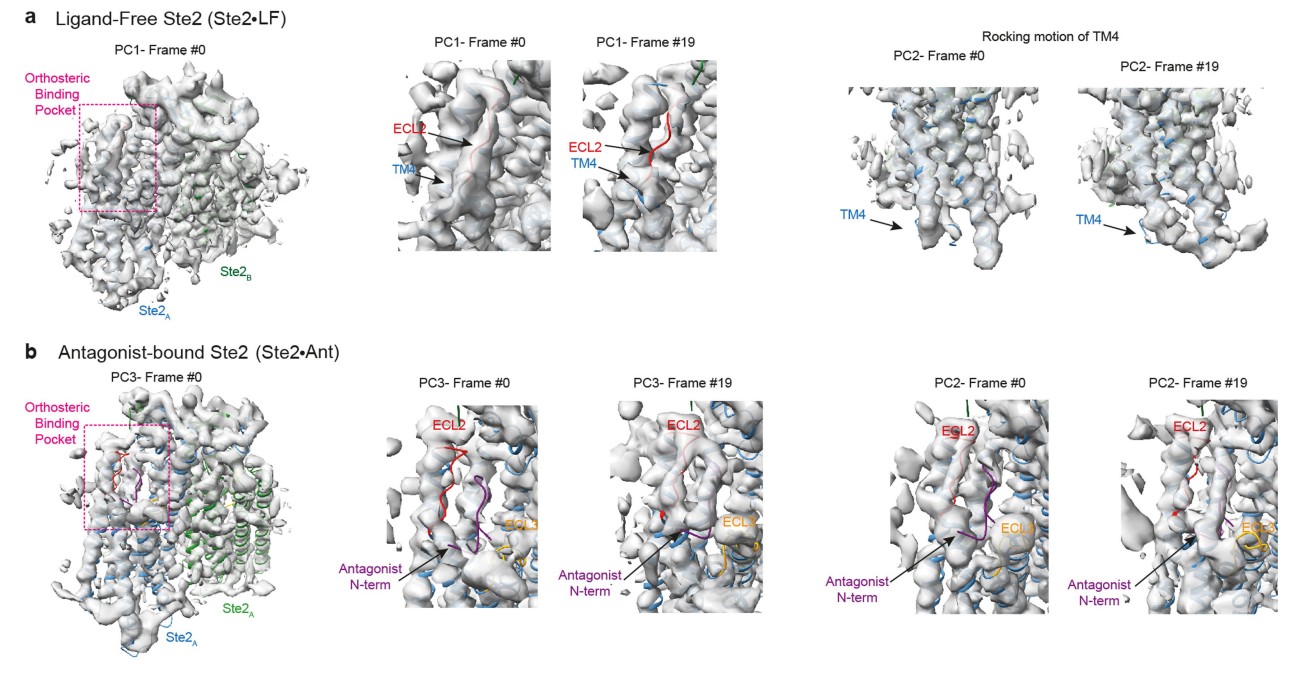

**a** Ligand-Free Ste2 (Ste2•LF)

**b** Antagonist-bound Ste2 (Ste2•Ant)

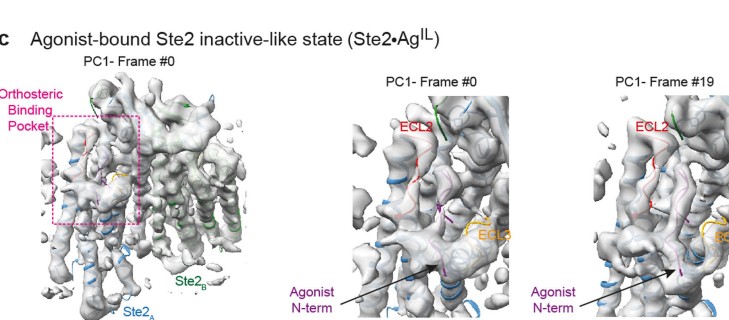

**c** Agonist-bound Ste2 inactive-like state (Ste2•Ag$^{IL}$)

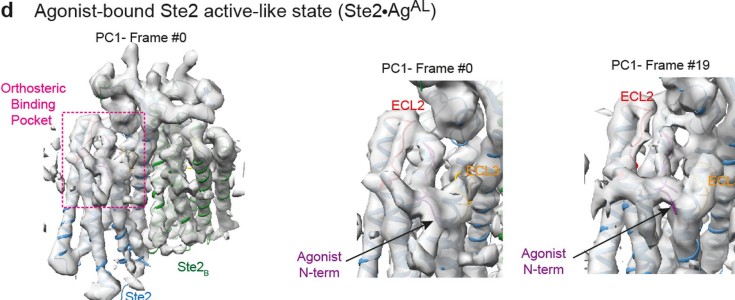

**d** Agonist-bound Ste2 active-like state (Ste2•Ag$^{AL}$)

**Extended Data Fig. 6 | 3D variability analysis of the cryo-EM data reveals different levels of flexibility in the engagement of the peptide in different activation states. a**, Frame #0 of PC1 reveals density for ECL2 but is absent in frame #19 of PC1. The greater degree of flexibility in ECL2 in the absence of bound peptide is consistent with its role in peptide binding as the side chains of residues T199$^{ECL2}$-F204$^{5x28}$ undergo flipping upon engagement with both the agonist and antagonist peptides (Extended Data Fig. 5). The flexible ECL2 is connected to TM4 that shows a rocking motion. **b**, Frame #0 of PC2 and PC3 reveals density for the C-terminal region of the antagonist, but no density for the N-terminal portion. Frame #19 of PC2 and PC3 reveals density for both the N-terminal and C-terminal regions. **c**, Comparison of start and end frames of

PC1 in Ste2$^{IL}$•Ag reveals strong density for both the N-terminal and C-terminal regions of α-factor, but higher conformational flexibility in the middle of the peptide. **d**, Comparison of start and end frames of PC1 in Ste2$^{AL}$•Ag reveals strong density for the entire α-factor indicating a stronger engagement with the receptor. Density maps are displayed as transparent surface representation (Ste2$_A$, blue; Ste2$_B$ green; ECL2, red; agonist (ago) and antagonist (ant) peptides, purple; ECL3, orange). The displayed density maps (surface representations) represent the indicated frames obtained from the 3D variability analysis procedure. The atomic model is from the respective cryo-EM structures that has been docked into the map frames for better visualization.

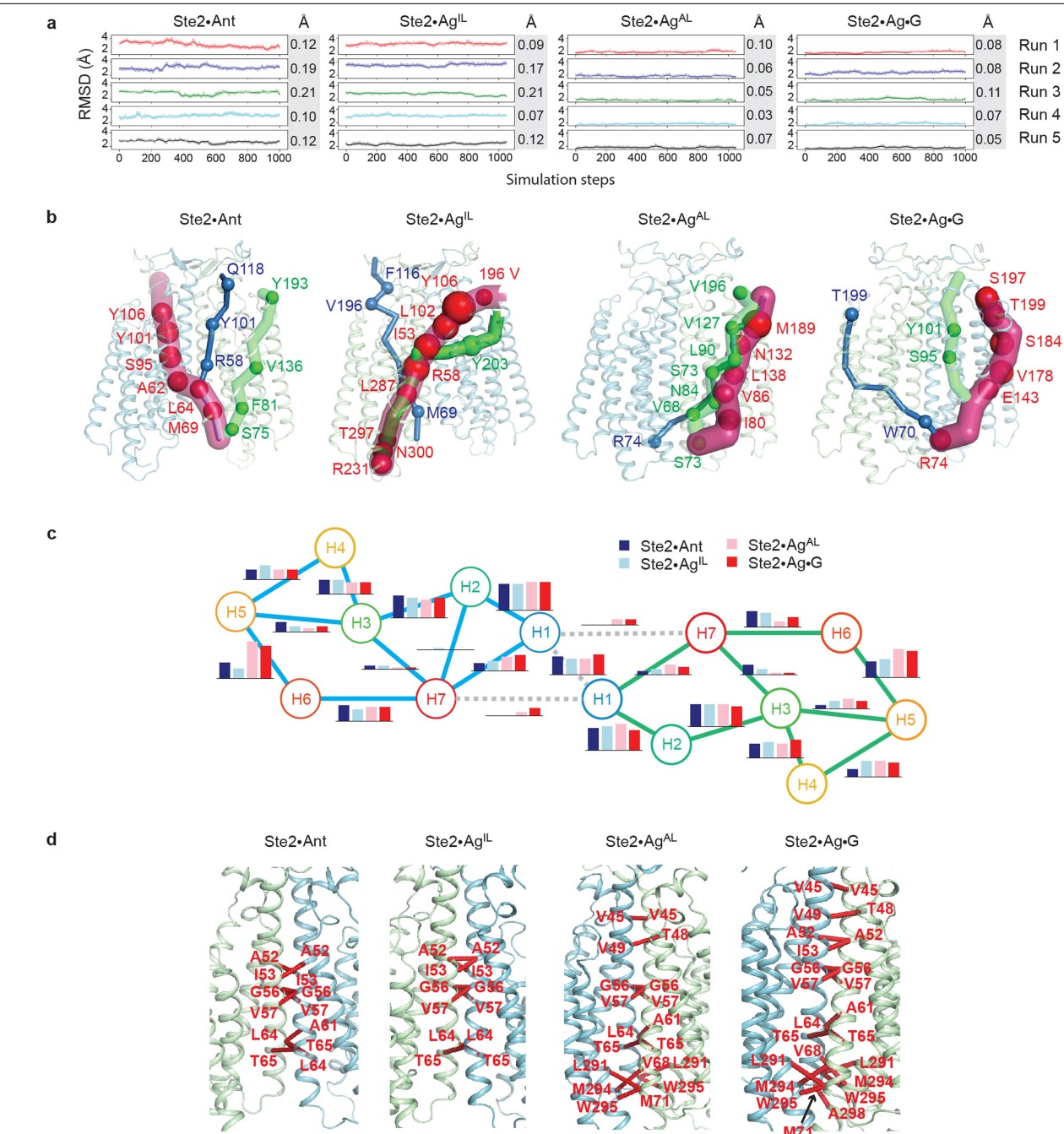

**Extended Data Fig. 7 | MD simulations, interhelical contacts and allosteric pipelines. a**, Root mean square deviation (RMSD) versus simulation steps for all MD simulation runs for each of the four conformational states. The RMSD was calculated for the Cα atoms of residues 30 to 301 over the trajectory from 200 ns to 400 ns. The moving average of the RMSD is shown as a solid line, with the fluctuations (in Å) given to the right of each panel suggesting that all simulations are equilibrated. **b**, The three top scoring allosteric communication pipelines of residues ranked by their strength of allosteric communication extending from extracellular loop regions to the G protein coupling site in all four conformational states: rank 1 (magenta), rank 2 (green),

rank 3 (blue). The strength of contribution of each residue is called the "hub score". Hub residues with hub score higher than ten are shown as spheres and labelled. Hub residues that show up in more than one allosteric communication pipeline are shown in the same colour as the top ranked pipeline and are not labelled twice. **d**, Bar graphs show the number of persistent contact (>60% simulation time) formed between each pair of TMs during MD simulation for Ste2·Ant (dark blue), Ste2[IL]·Ag (light blue), Ste2[AL]·Ag (pink) and Ste2·Ag·G (red). The Y-axis scale is 0–18 for all bar graphs. **e**, The persistent inter-residue contacts at the dimer interfaces are shown with red dash lines.

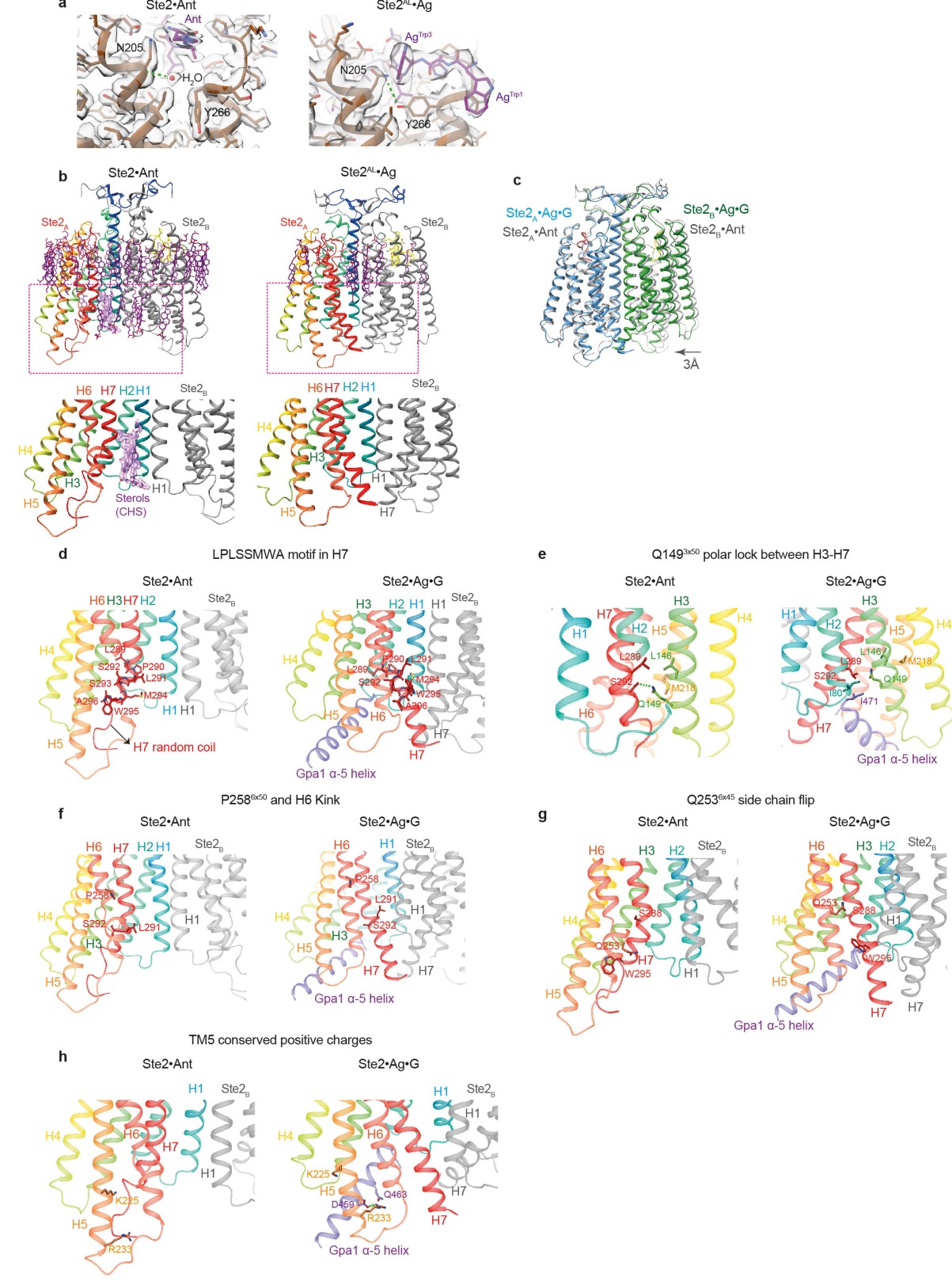

**Extended Data Fig. 8 |** See next page for caption.

**Extended Data Fig. 8 | Position of conserved residues in Ste2 conformational states. a**, In Ste2•Ant, a water molecule makes a polar interaction with the main chain of Asn205$^{5x29}$, whilst in the Ste2$^{AL}$•Ag state this site is occupied by Tyr266$^{6x58}$, which undergoes a rotamer change to form a polar interaction with the side chain of Asn205$^{5x29}$. The rotamer change in Tyr266$^{6x58}$ is triggered by its interaction with Trp1 of α-factor. Atomic models of Ste2 in the antagonist (map threshold 0.013) or agonist-bound active-like state (map threshold 0.560) are shown in pale brown, and the antagonist or agonist α-factor is shown as sticks in purple. **b**, Two putative sterol molecules, putatively assigned as CHS (purple), were found juxtaposed to H7 in the inactive state Ste2•Ant (map threshold 0.010) but there was no density in this region in the active state Ste2$^{AL}$•Ag (map threshold 0.560). Note that there were densities corresponding to sterols in other positions adjacent to the intracellular half of the receptor, but these were not modelled as the densities were not sufficiently resolved. **c**, On activation, changes in the dimer interface are accompanied by a relative shift of protomer Ste2$_B$ by ~3 Å towards the protomer Ste2$_A$. The models of Ste2•Ant, Ste2$_A$•Ag•G and Ste2$_B$•Ag•G are shown in grey, blue and green, respectively. **d**, The conserved LPLSSMWA motif (residues 289$^{7x49}$-296$^{7x56}$) on H7 is kinked inwards towards the receptor core in Ste2•Ant but forms an α-helix in Ste2•Ag•G. **e**, The inactive state (Ste2•Ant) is stabilized by a hydrogen bond formed between Q149$^{3x50}$ and S292$^{7x52}$. Upon activation (Ste2•Ag•G), Leu289$^{7x49}$ packs against Ile80$^{2x42}$ allowing Leu289$^{7x49}$ to interact with Ile471$^{H5.25}$ of the G protein α-subunit Gpa1. A hydrophobic core formed by Ile80$^{2x42}$, Leu289$^{7x49}$, and Ile471$^{H5.25}$ replace the polar lock present in the inactive state. **f**, The conserved Pro258$^{6x50}$ causes a kink in H6 in the inactive state. **g**, The side chain of Gln253$^{6x45}$ interacts with Trp295$^{7x55}$ of the LPLSSMWA motif in the inactive state, but undergoes a 180° flip upon activation as Gln253$^{6x45}$ moves towards Ser288$^{7x48}$ and Trp295$^{7x55}$ moves away to enable H7 to interact with H1 and H7 of the adjacent protomer. **h**, The conserved positively charged residues in TM5 (Lys225$^{5x49}$ and Arg233$^{5x57}$) undergo side chain changes upon activation to enable Gpa1 α5-helix binding and forms polar and ionic interactions with Gpa1.

**Extended Data Table 1 | Cryo-EM data collection and refinement statistics**

| | Ste2 Antagonist EMDB-13880 PDB 7QA8 EMPIAR-10877 | Ste2 Ligand-Free EMDB-13882 PDB 7QB9 EMPIAR-10878 | Ste2 Agonist[inactive-like] EMDB-13886 PDB 7QBC EMPIAR-10879 | Ste2 Agonist[active-like] EMDB-13887 PDB 7QBI EMPIAR-10879 |
|---|---|---|---|---|
| **Session** | **LMB Krios3** | **LMB Krios2** | **LMB Krios2** | **LMB Krios2** |
| **Data collection and processing** | | | | |
| Detector | K3 | Falcon4 | K2 | K2 |
| Magnification | 105,000x | 96,000x | 105,000x | 105,000x |
| Voltage (kV) | 300 | 300 | 300 | 300 |
| Electron exposure (e⁻/Å²) | 57 | 54 | 50 | 50 |
| Defocus range ($\mu$m) | -0.7 to -2.0 | -0.8 to -2.4 | -0.9 to -2.7 | -0.9 to -2.7 |
| Pixel size (Å) | 0.86 | 0.85 | 1.1 | 1.1 |
| Symmetry imposed | C2 | C2 | C2 | C2 |
| Initial particle images (no.) | 4,982,771 | 2,409,127 | 1,854,854 | 1,854,854 |
| Final particle images (no.) | 136,877 | 67,791 | 64,091 | 65,284 |
| Map resolution (Å) | 2.69 | 3.10 | 3.53 | 3.46 |
| FSC threshold | 0.143 | 0.143 | 0.143 | 0.143 |
| Map resolution range[†] (Å) | ~2.3 to ~5.9 | ~2.7 to ~5.2 | ~3.1 to ~5.4 | ~3.0 to ~5.4 |
| | | | | |
| **Refinement** | | | | |
| Initial model used (PDB code) | 7AD3 | 7AD3 | 7AD3 | 7AD3 |
| Model resolution[‡] (Å) | 2.9 | 3.3 | 3.8 | 3.7 |
| FSC threshold | 0.5 | 0.5 | 0.5 | 0.5 |
| Map sharpening $B$ factor (Å²) | -88.0 | -127.8 | -154.8 | -151.2 |
| Model composition | | | | |
| Non-hydrogen atoms | 6552 | 6253 | 5348 | 5224 |
| Protein residues | 622 | 598 | 622 | 622 |
| Ligands | 50 | 49 | 20 | 16 |
| $B$ factors (Å²) | | | | |
| Protein | 45.91 | 53.41 | 68.07 | 75.35 |
| Ligand | 56.87 | 61.28 | 66.75 | 69.68 |
| R.m.s. deviations | | | | |
| Bond lengths (Å) | 0.005 | 0.005 | 0.006 | 0.006 |
| Bond angles (º) | 0.777 | 0.820 | 0.965 | 0.871 |
| Validation | | | | |
| Molprobity score | 1.18 | 0.96 | 1.46 | 1.15 |
| Clashscore | 1.60 | 1.88 | 3.65 | 3.00 |
| Poor rotamers (%) | 0.37 | 0.19 | 0.00 | 0.19 |
| Ramachandran plot | | | | |
| Favored (%) | 96.07 | 97.98 | 95.60 | 97.71 |
| Allowed (%) | 3.93 | 2.02 | 4.40 | 2.29 |
| Disallowed (%) | 0.00 | 0.00 | 0.00 | 0.00 |

† Local resolution range.
‡ Resolution at which FSC between map and model is 0.5.

**Extended Data Table 2 | Amino acid residues with stubbed side chains in the atomic models**

| | N-Ter | TM1 | ICL1 | TM2 | ECL1 | TM3 | ICL2 | TM4 | ECL2 | TM5 | ICL3 | TM6 | ECL3 | TM7 |
|---|---|---|---|---|---|---|---|---|---|---|---|---|---|---|
| Ste2•LF | - | - | - | - | - | - | N158 | - | Q200<br>D201 | - | - | D242$^{6x34}$<br>M250$^{6x42}$ | - | - |
| Ste2•Ant | - | - | - | - | - | - | D157<br>N158 | - | - | - | - | M250$^{6x42}$ | - | - |
| Ste2$^{IL}$•Ag | D14<br>T35$^{D1S2x54}$ | - | R74<br>K77 | | - | I153$^{3x54}$<br>F154$^{3x55}$<br>T155$^{3x56}$ | D157<br>N158<br>F159<br>K160 | R161$^{4x41}$<br>I162$^{4x42}$<br>K187$^{4x67}$<br>N194$^{4x74}$ | Q200 | I227$^{5x51}$<br>L228$^{5x52}$<br>I230$^{5x54}$<br>R231$^{5x55}$<br>R233$^{5x57}$<br>R234$^{5x58}$<br>F235$^{5x59}$<br>L236$^{5x60}$ | L238<br>K239<br>Q240 | D242$^{6x34}$<br>H245$^{6x37}$<br>I246$^{6x38}$<br>I249$^{6x41}$<br>M250$^{6x42}$<br>S251$^{6x43}$<br>L256$^{6x48}$<br>I260$^{6x52}$<br>L264$^{6x56}$<br>S267$^{6x59}$ | - | T297$^{7x57}$<br>N301$^{7x61}$ |
| Ste2$^{AL}$•Ag | D14<br>Q21$^{D1S1x52}$<br>T35$^{D1S2x54}$ | - | K77 | | - | K151$^{3x52}$<br>V152$^{3x53}$<br>F154$^{3x55}$<br>T155$^{3x56}$ | D157<br>N158<br>F159<br>K160 | R161$^{4x41}$<br>I162$^{4x42}$<br>M165$^{4x45}$<br>K187-$^{4x67}$<br>N194$^{4x74}$ | Q200 | K202$^{5x26}$<br>F220$^{5x44}$<br>F221$^{5x45}$<br>I227$^{5x51}$<br>I230$^{5x54}$<br>R231$^{5x55}$<br>S232$^{5x56}$<br>R233$^{5x57}$<br>R234$^{5x58}$<br>F235$^{5x59}$<br>L236$^{5x60}$ | L238<br>K239<br>Q240<br>F241 | D242$^{6x34}$<br>S243$^{6x35}$<br>F244$^{6x36}$<br>H245$^{6x37}$<br>I246$^{6x38}$<br>Q253$^{6x45}$<br>L256$^{6x48}$<br>I260$^{6x52}$ | - | M294$^{7x54}$<br>N300$^{7x60}$<br>N301$^{7x61}$ |

# Reporting Summary

Nature Research wishes to improve the reproducibility of the work that we publish. This form provides structure for consistency and transparency in reporting. For further information on Nature Research policies, see our Editorial Policies and the Editorial Policy Checklist.

## Statistics

For all statistical analyses, confirm that the following items are present in the figure legend, table legend, main text, or Methods section.

| n/a | Confirmed | |
|---|---|---|
| ✗ | ☐ | The exact sample size (*n*) for each experimental group/condition, given as a discrete number and unit of measurement |
| ✗ | ☐ | A statement on whether measurements were taken from distinct samples or whether the same sample was measured repeatedly |
| ✗ | ☐ | The statistical test(s) used AND whether they are one- or two-sided<br>*Only common tests should be described solely by name; describe more complex techniques in the Methods section.* |
| ✗ | ☐ | A description of all covariates tested |
| ✗ | ☐ | A description of any assumptions or corrections, such as tests of normality and adjustment for multiple comparisons |
| ☐ | ✗ | A full description of the statistical parameters including central tendency (e.g. means) or other basic estimates (e.g. regression coefficient) AND variation (e.g. standard deviation) or associated estimates of uncertainty (e.g. confidence intervals) |
| ✗ | ☐ | For null hypothesis testing, the test statistic (e.g. *F*, *t*, *r*) with confidence intervals, effect sizes, degrees of freedom and *P* value noted<br>*Give P values as exact values whenever suitable.* |
| ✗ | ☐ | For Bayesian analysis, information on the choice of priors and Markov chain Monte Carlo settings |
| ✗ | ☐ | For hierarchical and complex designs, identification of the appropriate level for tests and full reporting of outcomes |
| ☐ | ✗ | Estimates of effect sizes (e.g. Cohen's *d*, Pearson's *r*), indicating how they were calculated |

*Our web collection on statistics for biologists contains articles on many of the points above.*

## Software and code

Policy information about availability of computer code

| Data collection | EPU 2.3.0.79 |
|---|---|
| Data analysis | AceDRG (via CCP-EM)<br>Allosteer v1.0<br>CCP-EM v1.3<br>CHARMM36-mar2019<br>CHARMM-GUI 3.0<br>Chimera 1.13.1<br>ChimeraX v1.2.5<br>Coot 0.9-pre EL<br>cryoSPARC v3.1<br>CTFFIND 4.1 (via RELION)<br>DeepEMHancer v1.0<br>eLBOW (via PHENIX)<br>g-cluster (via GROMACS)<br>get_contact (version 2021)<br>Gctf 1.18<br>GROMACS 2019.3<br>HOLLOW v1.3<br>LINCS (no version number defined)<br>Maestro protein-preparation wizard (Schrödinger Release 2021-2)<br>Molprobity (via PHENIX) |

Motioncor2 1.2.1
PHENIX 1.19.2-4158-000
phenix.resolve_cryo_em
Pymol 2.2.2
RELION 3.1
Warp 1.0.6

For manuscripts utilizing custom algorithms or software that are central to the research but not yet described in published literature, software must be made available to editors and reviewers. We strongly encourage code deposition in a community repository (e.g. GitHub). See the Nature Research guidelines for submitting code & software for further information.

## Data

Policy information about availability of data

All manuscripts must include a data availability statement. This statement should provide the following information, where applicable:

- Accession codes, unique identifiers, or web links for publicly available datasets
- A list of figures that have associated raw data
- A description of any restrictions on data availability

Structures have been deposited in the Protein Data Bank (PDB; https://www.rcsb.org/), and the associated cryo-EM data has been deposited in the Electron Microscopy Data Bank (EMD; https://www.ebi.ac.uk/pdbe/emdb/) and the Electron Microscopy Public Image Archive (EMPIAR; https://www.ebi.ac.uk/empiar/): Ste2•LF (PDB 7QB9, EMD-13882, EMPIAR-10878); Ste2•Ant (PDB 7QA8, EMD-13880, EMPIAR-10877); Ste2IL•ag (PDB 7QBC, EMD-13886, EMPIAR-10879); Ste2AL•ag (PDB 7QBI, EMD-13887, EMPIAR-10879). There are no restrictions on data availability.

# Field-specific reporting

Please select the one below that is the best fit for your research. If you are not sure, read the appropriate sections before making your selection.

[✗] Life sciences [ ] Behavioural & social sciences [ ] Ecological, evolutionary & environmental sciences

For a reference copy of the document with all sections, see nature.com/documents/nr-reporting-summary-flat.pdf

# Life sciences study design

All studies must disclose on these points even when the disclosure is negative.

| | |
|---|---|
| Sample size | Sample size was not predetermined. For the cryo-EM structure, the sample size used was sufficient because it produced a structure at sufficient resolution to identify side chains. For SEC and SDS-PAGE gels, experiments were performed twice or thrice as indicated and as there was no major differences between the experiments, the sample size was deemed sufficient. |
| Data exclusions | During the generation of the 3D cryo-EM structure, particles that did not align well with the major population (i.e. they were damaged, poor signal to noise, different conformation, lacking a subunit) were excluded from the data set. Inclusion of 'bad' particles would have had a detrimental effect on the overall resolution of the structure. No exclusions were made in biochemical assays. |
| Replication | Structure determination was performed once. Replication was not required because each structure represents the average structure of ca. 130,000 molecules for Ste2•Ant, 68,000 molecules for Ste2•LF, and ca. 65,000 molecules for Ste2IL•Ag and Ste2AL•Ag from data collections performed on different days with different input materials. A number of independent experiments were performed for biochemical assays and purifications as reported in the manuscript, and all attempts at replication were successful. |
| Randomization | This study did not allocate experimental groups thus no randomisation was required for the reported experiments. All variables could be well controlled. |
| Blinding | Experimental results were all quantitative and did not require group allocation or any subjective analysis, thus no experiments were performed with blinding |

# Reporting for specific materials, systems and methods

We require information from authors about some types of materials, experimental systems and methods used in many studies. Here, indicate whether each material, system or method listed is relevant to your study. If you are not sure if a list item applies to your research, read the appropriate section before selecting a response.

## Materials & experimental systems

| n/a | Involved in the study |
|---|---|
| ☒ | Antibodies |
| ☐ ☒ | Eukaryotic cell lines |
| ☒ | Palaeontology and archaeology |
| ☒ | Animals and other organisms |
| ☒ | Human research participants |
| ☒ | Clinical data |
| ☒ | Dual use research of concern |

## Methods

| n/a | Involved in the study |
|---|---|
| ☒ | ChIP-seq |
| ☒ | Flow cytometry |
| ☒ | MRI-based neuroimaging |

# Eukaryotic cell lines

Policy information about cell lines

| | |
|---|---|
| Cell line source(s) | Trichoplusia ni (Expressions Systems & Thermo Fisher) |
| Authentication | The cell lines were not authenticated by the authors. The supplier maintained the cell line and did not specify means of authentication. |
| Mycoplasma contamination | The cell lines were not tested by the authors for mycoplasma contamination as this was performed by the supplier. |
| Commonly misidentified lines (See ICLAC register) | No commonly misidentified cell lines were used |

