## [Peer Review File · Nature]

Manuscript Title: Activation mechanism of the Class D fungal GPCR dimer Ste2

Reviewer Comments & Author Rebuttals

Reviewer Reports on the Initial Version:

Referee expertise:

Referee #1: GPCR structure/function, cryoEM

Referee #2: Class D receptors

Referee #3: Computational

Referees' comments:

Referee #1 (Remarks to the Author):

This manuscript describes structures of the *S. cerevisiae* Ste2, a dimeric receptor which belongs to the Family D of GPCRs found exclusively in fungi. The authors recently published a structure of the Ste2 dimer coupled to two heterotrimeric yeast G proteins, with notable differences in the intracellular positioning of TM4 and the mode of G protein engagement compared to other Family A, B and F GPCRs. In the present study, the authors continued their structural characterization of Ste2 and show the structure of the receptor in the absence of ligand (apo), with inverse agonist bound, and with native agonist bound. The inactive state structure is novel, and one of the active state structures is similar but not identical to the previously determined structure. Comparison of these structures with the previously determined G protein-coupled Ste2 (a total of 5 cryo-EM structures) enables the authors to describe more or less in full the necessary molecular rearrangements underlying the activation mechanism of this Class D GPCR. As had already been elucidated from their previous study with the G protein complex, the activation mechanism of this class is significantly different from Family A, B and F GPCRs.

Overall, the authors seem to have done a really great job in producing high quality cryo-EM maps and most of the modeling seems robust (see some specific concerns below). The manuscript is mostly well written and clear. Furthermore, the MD simulations provide a really nice addition to the work as they provide insights to the communication between the two protomers through the dynamic dimer interface. It is an intriguing result that the strongest motion allosteric networks run through the dimer interface in the inactive states, while in the active states the strongest correlated motions run within each protomer.

Comments on modeling:

Although most interpretations in the manuscript are straightforward and well supported, I find some of the modeling and associated discussion problematic. In general, I would suggest that the authors carefully check all regions, particularly the loops, and only model what is justified by the density. Where side chains are not visible, the residues should be stubbed.

As specific examples:

The "Ligand Free" map around Phe204 is not well resolved but has been modeled as in the high-resolution "Antagonist" map, and thus the interpretations for this region are not fully justified for the Ste2•LF structure. Accordingly, Extended Data Fig. 5b may be partly inaccurate. Most of the sidechains on the preceding loop should be stubbed since there is no associated density. Many of the other loops were also modeled with sidechains without sufficient density. I note that the

results of mutating Phe204 and N205 shows the importance of this region for ligand binding but not for the supposed conformational change.

The authors state that "Upon antagonist binding, the side chains of Asp201 and Phe204 flip inwards by 180°", however, there is no density in the Ste2•LF for the Asp201 sidechain. Similarly, they state that "Lys202 flips in the opposite direction as it clashes with the ligand", although there is no clear density for Lys202 in the high-resolution Ste2•Ant map. I would suggest that the authors only describe the flexibility of this region in the absence of a ligand and probably the interaction of Phe204 with the ligands.

The EM density for helices 5, 6, and 7 in the active like agonist bound structure (Ste2AL•Ag) is poor and the model is not justified for these regions. Most side chains should be stubbed. I would suggest the authors clearly state that this region is not well resolved, likely because it samples a range between inactive and active state conformations in the absence of G protein.

All lipidic like densities have been modeled as CHS although some of these densities are definitely not sufficient to model anything and most others are not definitive. As the authors state, some of these densities could be other sterols. I also note that GDN was used in solubilization, which has a similar ring structure to CHS. It would thus be safest to remove most of the CHS molecules from the deposited model.

Other comments:

How confident are the authors about the C2 symmetry approach for their reconstructions? Recent Family C receptor structures showed asymmetric transmembrane interfaces, where small deviation in symmetry may be only visible at high resolution reconstructions. It would be helpful if the authors discuss the similarity of C1 symmetry reconstruction with the final C2 symmetry reconstructions. Related to this, two-thirds of the particles at the last 3D classification of the active state dataset seem to yield several heterogeneous but stable reconstructions. Have the authors explored potential asymmetry in these states?

Even though I agree that a full exploration of the communication at the dimer interface merits a separate study of its own, it would be helpful to show through MD simulations just a couple of selected mutants of residues identified by both mutagenesis and the existing simulations, and which would disrupt intradimer communication. Such results would go a long way in supporting the current findings and the computational approach.

Editorial comments:

The authors discuss P2907x50 as a conserved residue on Ste2 that facilitates a formation of kink on helix7 contributing to the activation mechanism, in contrast to Family A GPCRs. Please include a discussion on a similar kink observed in some Family B GPCRs (PDB IDs; 5UZ7, 7KNT).

Local resolution maps are shown with a very wide resolution range, resulting in an insufficient color distribution to represent the protein densities. The lower resolution end should be limited to a smaller number for each map, thereby allowing better color distribution to represent the resolution variation across the protein densities.

Some figures may merit from clearer illustration. For example, I find the representation of structure and contacts in Fig 2d confusing. Also, the presence of all the CHS molecules on the ribbon representation of Fig 1 makes hard to see the underlying receptor structure.

The stabilization of the inactive state for membrane extraction through the addition of G protein is nifty. Nevertheless, it does depend on the formation of some kind of pre-coupled state which may or may not be widely applicable, especially for the monomeric GPCRs. In the absence of such demonstration, perhaps through a methodology targeted study, I am not sure that the use of an acronym (PSGWAY) to describe the method is very useful at this point.

The structures shown in Figure 5 should be referenced in addition to mentioning their PDB IDs.

Referee #2 (Remarks to the Author):

The manuscript by Velazhahan et al. presents an important new structural view of a receptor that has served as a model for much understanding of G protein coupled receptor signaling. The four

presented structures, one ligand free, one with antagonist, and two with bound agonist, appear to be carefully determined and, in the case of the ligand-free state, required a new approach for receptor purification that is expected to be applicable to other receptors. The work provides a new view of the growing diversity of mechanisms by which different classes of GPCRs can interact with different types of ligands and activate different G proteins and of the multiple steps that can be involved in transitions between inactive and active states. It also provides new insights into the way in which receptor-receptor interactions in a receptor dimer can be involved in signaling mechanisms and into the role of interactions of receptors with steroids. The described work comprises an important structural framework for understanding diverse mutational, biochemical, and biophysical approaches that have been applied to understanding pheromone signaling in yeast over the past several decades. Thus, I strongly support publication in Nature, subject to minor revisions to address the points raised below:

1. The manuscript states, "Ste2 is a homodimer that can couple to two G proteins simultaneously." While there is evidence that each receptor must signal through its associated G protein, it is not clear that this happens simultaneously, especially since the previous structure of the dimeric Ste2-G protein complex shows significant asymmetry with respect to receptor-G protein interactions.

2. The cited references do not support the manuscript's characterization of the "antagonist" [desTrp1Ala3Nle12] α -factor as an "inverse agonist". This ligand is not known to inhibit basal activity of the pheromone pathway and several publications show that it, in fact, exhibits partial agonist activity toward constitutively active mutant receptors.

2. I think it would be helpful, from the beginning of the manuscript to state that the current structures are of receptor alone in the presence of ligands, whereas the previous structure was of a ligand-bound receptor-G protein complex.

3. It would be informative to include an additional panel in Fig. 1 showing the ligand conformation in the Ste2-Ag-G complex for comparison to panels 1e-g (as was done in many of the other figures in the paper.).

4. The cited referenced (#10) regarding the kinetics of ligand binding supports the existence of two kinetic phases, but does not indicate the nature of those phases. Initial binding of the C-terminal of the ligand is discussed in a number of other publications.

5. In Fig. 4b, there is no vertical scale for the numbers of contacts in the bar graphs.

6. It would be interesting to know the nature of the gel filtration peak to the left of the main peak in Extended Data Fig. 1. Is there a gel lane of this peak or EM characterization? Why would it be absent for the antagonist sample?

7. Can anything be concluded from the small percentages of imaged particles used for the reconstructions? Could the large majority of particles not used be present in some dominant state that was missed in the analysis? This is of particular concern for the ligand-free receptors, which were purified in the presence of G protein to maintain stability, but are then analyzed following removal of the G protein. This is partly addressed in the case of the "heterogeneous" states of the Ste2-Ag sample. Could such heterogeneous states be present for the other examined samples? It would be particularly interesting to know whether non-dimeric particles were observed in any of the samples.

8. The manuscript would benefit from some additional explanation of the "variability" analysis for the benefit of non-specialists. The procedure is apparently fairly recently described. It seems to model variability of different regions in the cryo-EM reconstructions. However, the manuscript seems to imply that the procedure generates a temporal series of frames describing ligand binding. It is not clear how this would be derived. It is also not clear what validation is available for

this approach, either in general or for the analysis specifically of the Ste2 samples. Also, it seems that “variability” and “variance” are used interchangeably in the text, which seems confusing. In Extended Data Fig. 6, it would seem to be important to indicate exactly what structures the displayed peptide chains, as opposed to the surface representations, correspond to. Is one the static consensus structure? The two representations are clearly not from the same structures, since they don’t overlap. Why are different PC components displayed for the different samples? Why is there no PC1 from the antagonist sample? How was it decided how many PC’s to display in each case?

9. Extended Data Fig. 4 shows modeling only of the helical segments in the structures. Given the importance of loop regions, and the apparent poorer fit to the data at the helix extremities, it would seem to be important to provide a similar Extended Data figure for the loops. The poorer modeling does not necessarily seem to correlate with regions of lower resolution shown in Extended Data Fig. 3.

10. If C2 symmetry was applied to the reconstructions, why are the ligand binding pocket volumes of the two monomers different for each structure shown in Extended Data Fig. 5e?

11. It is stated in the text that, “The flattening of the RMSD curves suggest (sic) that all simulations are equilibrated. However, such flattening with simulation time does not seem apparent in Extended Data Fig. 5a. If there is not flattening, what does this imply about the simulations?

12. There are many reports that Ste2p can activate mammalian or yeast-mammalian chimeras. How does this fit with the proposed specific mechanism of class D receptor activation?

13. The manuscript seems to rely heavily on molecular dynamics calculations to address points that may be directly addressable from the structural data. Given the quality of the structural data showing changes in dimer contacts in different states, it seems gratuitous to then focus on residues that remain in contact 60% of the time in MD simulations. A similar argument could be made for allosteric changes in the receptor, given that the particular changes in the different states can be directly observed.

Referee #3 (Remarks to the Author):

A.Summary of the key results: Done.

B.Originality and significance: Novel and corresponding references are included in the manuscript.

C.Data & methodology: The methods followed in running molecular dynamics simulations and analyses are appropriate.

D. Appropriate use of statistics and treatment of uncertainties: Error bars are provided in presenting dimer interaction energies.

E. Conclusions are robust and reliable.

F. a. The name of the force field used and the ionization state of His residue found in the α -factor should be given. b.In Extended Figure 7b, allosteric interaction networks are given for different systems. It can be understood that one can get different pathways for antagonist-bound receptor;however, it is interesting to observe that pathways differ in the G protein-bound system. It would be informative if the authors can comment on it.

G.Corresponding references are provided in the manuscript.

H.The manuscript is well-written and logically designed.

Author Rebuttals to Initial Comments:

Reply to Referees' comments

We thank the reviewers for their constructive comments. We have addressed all the comments in the revised manuscript as described below.

Referee #1 (Remarks to the Author):

1. Although most interpretations in the manuscript are straightforward and well supported, I find some of the modeling and associated discussion problematic. In general, I would suggest that the authors carefully check all regions, particularly the loops, and only model what is justified by the density. Where side chains are not visible, the residues should be stubbed.

In the intermediate active-like Ste2^{AL}•Ag and inactive-like state Ste2^L•Ag structures, side chains are not well resolved for several residues in the intracellular loops 2 and 3 (ICL2 and ICL3, respectively) and the adjacent regions at the end of the helices H3-H6. Initially, these were either stubbed or modelled based on the high-resolution active Ste2•Ag•G or inactive Ste2•Ant state structures. We note the concern of the reviewer and have now stubbed all these residues and have provided a table of all stubbed residues, including those that have been stubbed in the original models (Extended Data Table 2).

2. The “Ligand Free” map around Phe204 is not well resolved but has been modeled as in the high-resolution “Antagonist” map, and thus the interpretations for this region are not fully justified for the Ste2•LF structure. Accordingly, Extended Data Fig. 5b may be partly inaccurate. Most of the sidechains on the preceding loop should be stubbed since there is no associated density. Many of the other loops were also modeled with sidechains without sufficient density. I note that the results of mutating Phe204 and N205 shows the importance of this region for ligand binding but not for the supposed conformational change. The authors state that “Upon antagonist binding, the side chains of Asp201 and Phe204 flip inwards by 180°”, however, there is no density in the Ste2•LF for the Asp201 sidechain. Similarly, they state that “Lys202 flips in the opposite direction as it clashes with the ligand”, although there is no clear density for Lys202 in the high-resolution Ste2•Ant map. I would suggest that the authors only describe the flexibility of this region in the absence of a ligand and probably the interaction of Phe204 with the ligands.

We agree that there is no clear density for the whole of the side chain of Asp201, but there is sufficient to see that it has to flip (Extended Data Fig. 3i). However, given the ambiguity, we have stubbed the side chain and removed discussion from the main text. The case for Lys202 is different because there is visible density for the side chain in Ste2•Ant if the contouring is changed and the map sharpened. (Extended data Fig. 3g). In addition, there is no possibility of the Lys202 side chain remaining in the position it adopts in Ste2•LF once an antagonist or agonist binds because it would clash with the ligand. Given these two points, we have not removed the discussion about Lys202, but pointed out the density is weaker in Ste•Ant as it projects into the detergent micelle. The amended text is (lines 90-93):

‘Upon antagonist binding, the side chains of Phe204^{5x28} flips inwards by 180° (7 Å inward shift of the C α atom). Lys202^{5x26} has to flip in the opposite direction to prevent it clashing with the ligand (Extended Data Fig. 5b), although there is only weak density for the side chain in Ste2•Ant because it extends into the detergent micelle (Extended Data Fig. 3g,h).’

3. The EM density for helices 5, 6, and 7 in the active like agonist bound structure (Ste2AL•Ag) is poor and the model is not justified for these regions. Most side chains should be stubbed. I would suggest the authors clearly state that this region is not well resolved, likely because it samples a range between inactive and active state conformations in the absence of G protein.

We agree that the densities for helices 5, 6 and 7 in Ste2^{AL}•Ag, particularly at the ends of the helices, is not as good as that seen for the rest of the receptor. However, local resolution of helices 5-7, with the exception of ICL2 and ICL3 and residues proximal to these loops, still reached 3.4-3.6 Å and modelling of majority of the residues is possible by visualization after careful adjustment of B-factor map sharpening (Extended Data Fig. 3c,d). We have also now provided maps generated by the density modification procedure in Phenix (phenix.resolve_cryo_em) with only the unfiltered half-maps and molecular weight as inputs (see methods) that allows better visualization of some side chains (see attached files added to submission). Thus, with the exception of the stubbed residues in and around ICL2 and ICL3 that exhibit high flexibility, densities for the majority of the residues in transmembrane helices 5, 6 and 7 were visible in the Ste2^{AL}•Ag structure (see Extended Data Fig. 4c,d). For example, densities were present for the TM5 residues D201^{5x25} to A229^{5x53} (except K202^{5x26}, S219^{5x43}, F220^{5x44}, and I227^{5x51}), TM6 residues L247^{6x39} to L268^{6x60} (except Q253^{6x45}, I256^{6x48} and I260^{6x52}) and TM7 residues V276^{7x36} to A299^{7x59} (except M294^{7x54}). All residues for which side chains could not be assigned have been stubbed (see Extended Data Table S2). To explicitly state the weaker densities in the flexible regions of the map, we have clarified the main text to read (lines 170-173):

'The cryo-EM density for Ste2^{IL}•Ag and Ste2^{AL}•Ag shows lower resolution in H5-H7 than observed in Ste2•Ant and Ste2•Ag•G, resulting in only a proportion of the side chains being modelled (see Extended Data Table 2), which highlights their dynamic role in the conformational change.'

4. All lipidic like densities have been modeled as CHS although some of these densities are definitely not sufficient to model anything and most others are not definitive. As the authors state, some of these densities could be other sterols. I also note that GDN was used in solubilization, which has a similar ring structure to CHS. It would thus be safest to remove most of the CHS molecules from the deposited model.

As we described in the text, we had assigned all sterol-like densities putatively as CHS since CHS was present in vast molar excess during purification of the receptor. GDN has two additional ring structures in the hydrophobic region compared to CHS, making it significantly larger than CHS, so it seems very unlikely that the densities represent this detergent. However, we agree that we cannot definitively assign the densities as CHS, hence we used the term 'putative' throughout the text. We have removed several CHS molecules from the original models where EM densities were relatively weaker and have retained CHS molecules in regions where sterol-like densities could be reasonably visualized. We note that the succinate portion may not be clearly visible owing to its flexibility. We have clarified the text to read (lines 74-79):

All the structures contained densities on the periphery of the transmembrane regions of Ste2 that were attributed to putative sterols (Fig. 1a-d and Extended Data Fig. 4g). They were either unmodelled or, where the density was sufficiently strong, assigned putatively as cholesterol hemisuccinate (CHS) as this was present in vast molar excess throughout receptor purification; we cannot exclude the possibility that the densities may represent other sterols.

5. How confident are the authors about the C2 symmetry approach for their reconstructions? Recent Family C receptor structures showed asymmetric transmembrane interfaces, where small deviation in symmetry may be only visible at high resolution reconstructions. It would be helpful if the authors discuss the similarity of C1 symmetry reconstruction with the final C2 symmetry reconstructions.

We performed the majority of the data processing and classifications in C1 symmetry (as described in the methods) and we only applied C2 symmetry to the final reconstructions. This was because the maps obtained from C1 symmetry and C2 symmetry were virtually identical and application of C2 symmetry improved the resolution of the maps by ~ 0.2-0.3 Å, which is expected to occur only if the two protomers are identical. High resolution information was present in all the maps obtained with C1 symmetry and indicated no significant differences between the two protomers. For these reasons, we applied C2

symmetry to further improve the resolution of the maps and increase confidence in model building particularly of the Ste2^{IL}•Ag and Ste2^{AL}•Ag structures.

Our previously published G protein-coupled Ste2 structure (Ste2•Ag•G) structure was refined in C1 symmetry and there was also no significant differences observed between the two Ste2 protomers, despite disorder in one of the coupled G proteins. However, the disorder in one of the G proteins probably arose through interactions between the two coupled G proteins and not due to asymmetry in the receptor.

The structures of Ste2 in the inactive state and Ste2•Ag•G are both at 'high-resolution' and we see no evidence of differences between the protomers in the respective dimers. We are therefore confident in our use of C2 symmetry.

In contrast to what we see with Ste2, there is clear asymmetry in the transmembrane helices as well as large differences between the protomers visible in the recently reported Class C GPCR structures mGlu2/2 (PDB 7E9G), mGlu2/2 (PDB 7MTS) and mGlu4/4 (PDB 7E9H) coupled to a G protein heterotrimer. These structures were reported when our manuscript was under consideration. In Class C GPCRs, asymmetry appears to be critical as only one G protein heterotrimer couples to one protomer of the receptor dimer, and where present allosteric small molecules preferentially bind only to one of the two protomers. On the other hand, as shown in our previous work, each protomer of the Ste2 dimers is capable of binding to one G protein heterotrimer and one agonist peptide, and hence the symmetric Class D GPCR dimer Ste2 binds two G proteins simultaneously.

We have clarified the methods section to read (lines 665-671):

'C2 symmetry was applied to the final reconstruction, after 3D classifications, since no significant differences between the two protomers were observed in the high-resolution cryo-EM maps obtained in C1 symmetry. In addition, application of C2 symmetry improved the resolution of the maps by ~0.2-0.3 Å, which is expected to occur only if the two protomers are identical. The improved map resolution increased confidence in model building in regions with relatively lower local resolution in the intermediate state Ste2^{IL}•Ag and Ste2^{AL}•Ag structures.'

We have also included a discussion of the asymmetry recently reported for Class C GPCRs in the main text to read (lines 310-312):

'In addition, Class C GPCR dimers are asymmetric and couple to a single G protein³², in contrast to the symmetric dimer of Ste2 that can couple simultaneously to two G proteins¹.

6. Related to this, two-thirds of the particles at the last 3D classification of the active state dataset seem to yield several heterogeneous but stable reconstructions. Have the authors explored potential asymmetry in these states?

The heterogeneous classes obtained during the processing of Ste2^{AL}•Ag dataset did not show any evidence for asymmetry as similar heterogeneity was visible in both the protomers of the dimer.

7. Even though I agree that a full exploration of the communication at the dimer interface merits a separate study of its own, it would be helpful to show through MD simulations just a couple of selected mutants of residues identified by both mutagenesis and the existing simulations, and which would disrupt intradimer communication. Such results would go a long way in supporting the current findings and the computational approach.

We agree that it would be nice to quickly show a couple of mutations that disrupt communication across of the dimer interface. However, it is challenging to predict how a mutant would affect or divert the allosteric communication, and in previous work on other GPCRs changing a single amino acid residue

to different residues can have diametrically opposed effects. Effects may also be subtle and, rather than just block communication, the pipeline may shift through other residues. Therefore, we will (as the referee suggests) do a proper, rigorous study which will need considerable resources to complete both the computational aspects and also validation through rigorous experimentation. As we already pointed out (lines 235-239), mutations in residues predicted to be in the allosteric communication pipeline are known to affect ligand binding and/or receptor activation. It is also worth pointing out that this is not a new technique to identify allosteric pipelines in receptors and has been extensively validated on other GPCRs (references 18, 19, 71-73 in the manuscript).

8. The authors discuss P2907x50 as a conserved residue on Ste2 that facilitates a formation of kink on helix7 contributing to the activation mechanism, in contrast to Family A GPCRs. Please include a discussion on a similar kink observed in some Family B GPCRs (PDB IDs; 5UZ7, 7KNT).

It is interesting that both Class B and Class D1 receptors utilise a proline residue to allow a helix to kink as part of the mechanism of activation. We have included a brief discussion of this (lines 257-261) but unfortunately a longer discussion is not possible due to space constraints.

'The 47° kink at Pro290^{7x50} in the inactive state of Ste2 is reminiscent of the 60° kink in H6 of Class B GPCRs at the conserved sequence PxxG^{6.50b} that is essential for the outward movement of its cytoplasmic end to form the G protein binding cleft during receptor activation²¹. However, the sequence motifs around the respective Pro residues are different, as are the mechanisms of activation of Class B and Class D1 GPCRs (see Discussion).'

9. Local resolution maps are shown with a very wide resolution range, resulting in an insufficient color distribution to represent the protein densities. The lower resolution end should be limited to a smaller number for each map, thereby allowing better color distribution to represent the resolution variation across the protein densities.

Local resolution maps (Extended Data Fig.3) are now shown with a narrower range and smaller lower resolution number to highlight the variation across resolution.

Some figures may merit from clearer illustration. For example, I find the representation of structure and contacts in Fig 2d confusing.

Fig2d has been amended to give a clearer representation of the packing of helices around H7.

Also, the presence of all the CHS molecules on the ribbon representation of Fig 1 makes hard to see the underlying receptor structure.

CHS molecules have been removed from the ribbon representation of Fig.1 for better clarity of the underlying receptor structure.

The stabilization of the inactive state for membrane extraction through the addition of G protein is nifty. Nevertheless, it does depend on the formation of some kind of pre-coupled state which may or may not be widely applicable, especially for the monomeric GPCRs. In the absence of such demonstration, perhaps through a methodology targeted study, I am not sure that the use of an acronym (PSGWAY) to describe the method is very useful at this point.

The acronym PSGWAY (Pre-Stabilization of a GPCR by Weak Association) directly stands for the principle behind the technique, i.e. pre-stabilization of the GPCR before extraction from lipid membranes by using a G protein that forms a weakly associating complex in the absence of any bound ligands. As discussed in the methods section, a similar strategy exploiting the formation of a transient G protein complex has been shown to allow purification of CGRP receptor in the ligand-free state, which is a

Class B1 monomeric GPCR. However, the PSGWAY technique uses a different methodological framework that involves incubation with purified G proteins, instead of co-expression of a modified G_{α11} G protein, to achieve purification of the ligand-free state of the GPCR and could be more readily adopted since no additional changes to the expression conditions are necessary. We rationalized that giving the methodology an acronym that directly reflects the principle behind the technique would increase its visibility and help the GPCR community to explore adaptation of the strategy to facilitate purification of other GPCRs in the ligand-free states for structural studies. This is important because purification of GPCRs in the ligand-free state is usually very difficult owing to their instability in the absence of ligands. We would therefore like to keep the acronym in the text.

The structures shown in Figure 5 should be referenced in addition to mentioning their PDB IDs.

All the structures are now referenced.

Referee #2 (Remarks to the Author):

1. The manuscript states, “Ste2 is a homodimer that can couple to two G proteins simultaneously.” While there is evidence that each receptor must signal through its associated G protein, it is not clear that this happens simultaneously, especially since the previous structure of the dimeric Ste2-G protein complex shows significant asymmetry with respect to receptor-G protein interactions.

We agree entirely with the referee that signalling from the Ste2 dimer may not be a simultaneous process, which is why we did not write this. We wrote ‘Ste2 is a homodimer that can couple to two G proteins simultaneously...’ which is an entirely accurate statement as signalling and coupling are two different events. It seems superfluous to also state what the dimer might not do in the absence of any evidence so we would prefer not to mention anything about signalling.

2. The cited references do not support the manuscript’s characterization of the “antagonist” [desTrp1Ala3Nle12]α-factor as an “inverse agonist”. This ligand is not known to inhibit basal activity of the pheromone pathway and several publications show that it, in fact, exhibits partial agonist activity toward constitutively active mutant receptors.

We have removed the term “inverse agonist” and instead refer to the peptide [desTrp1Ala3Nle12]α-factor as “antagonist”.

2. I think it would be helpful, from the beginning of the manuscript to state that the current structures are of receptor alone in the presence of ligands, whereas the previous structure was of a ligand-bound receptor-G protein complex.

We have added this to the Abstract (lines 31-33):

‘We therefore determined structures of Ste2, in the absence of G protein, in two different conformations bound to the native agonist α-factor, bound to an antagonist or without any ligand.’

3. It would be informative to include an additional panel in Fig. 1 showing the ligand conformation in the Ste2-Ag-G complex for comparison to panels 1e-g (as was done in many of the other figures in the paper.).

Figure 1h has been added to show the ligand conformation in the Ste2•Ag•G protein structure.

4. The cited referenced (#10) regarding the kinetics of ligand binding supports the existence of two kinetic phases, but does not indicate the nature of those phases. Initial binding of the C-terminal of the ligand is discussed in a number of other publications.

That is indeed true and there are a number of publications that address how α -factor binds. On reflection, we feel it is better to cite a review on the whole topic as each primary paper adds a facet to the story and the review does a good job in pulling these strands together: 'Naider, F. & Becker, J. M. A Paradigm for Peptide Hormone-GPCR Analyses. *Molecules* **25**, (2020) 10.3390/molecules25184272'

5. In Fig. 4b, there is no vertical scale for the numbers of contacts in the bar graphs.

A vertical scale has been added

6. It would be interesting to know the nature of the gel filtration peak to the left of the main peak in Extended Data Fig. 1. Is there a gel lane of this peak or EM characterization? Why would it be absent for the antagonist sample?

The chromatograms shown in Extended Data Fig. 1 are from the final size-exclusion purification step that was performed to obtain a pure sample for cryo-EM studies. The peak to the left of the main peak in Extended Data Fig. 1 represents an 'aggregate peak' that is commonly observed during the purification of membrane proteins and runs in the void volume of the column. It consists of some aggregated receptor and primarily other contaminating proteins. We have now clearly labelled this peak as 'void'. The aggregate peak is present, but to a slightly lower extent, in the case of the antagonist-bound Ste2 sample and it simply reflects the relative homogeneity of the sample before the size-exclusion purification run.

7. Can anything be concluded from the small percentages of imaged particles used for the reconstructions? Could the large majority of particles not used be present in some dominant state that was missed in the analysis? This is of particular concern for the ligand-free receptors, which were purified in the presence of G protein to maintain stability, but are then analyzed following removal of the G protein. This is partly addressed in the case of the "heterogeneous" states of the Ste2-Ag sample. Could such heterogeneous states be present for the other examined samples? It would be particularly interesting to know whether non-dimeric particles were observed in any of the samples.

In all cryo-EM structures, only a small percentage of imaged particles contribute to the final high-resolution reconstructions since other particles are either heterogenous or are damaged due to interactions at the air-water interface during cryo-EM specimen preparation, and hence are classified 'out' in the 3D image classification procedures. It is typical that out of 5-10 million particles picked in a typical cryo-EM dataset, only ~100,000 to 300,000 particles contribute to the final 3D reconstruction.

In the case of the ligand-free Ste2 cryo-EM dataset, only 463,024 particles that resulted from 3D classifications of the initially picked 2,409,127 particles contained features consistent with intact transmembrane helices, as other particles were presumably damaged at the air-water interface or were empty micelles. This class of 463,024 particles were further classified into five classes that did not show any significant differences in the conformations between the classes, unlike the case with the Ste2^{AL}•Ag dataset where particles contributed to two distinct conformations of the receptor (Ste2^{AL}•Ag and Ste2^{IL}•Ag states) and additional classes with clear heterogenous conformations were visible. The final set of 67,791 particles represents particles with the highest resolution information and contributed to the final reconstruction. The discarded particles may have been partially damaged and/or include particles with non-favourable contrast transfer function (CTF) parameters indicating loss of information in the highest spatial frequencies.

In all of the Ste2 samples and corresponding datasets, no classes consistent with a non-dimeric receptor was observed despite extensive 2D and 3D classification procedures. We have added the following text (lines 631-633):

'These particles were consistent with a homodimeric Ste2 in all three datasets and no reliable classes consistent with a Ste2 monomer were obtained despite extensive 2D and 3D classifications.'

8. The manuscript would benefit from some additional explanation of the “variability” analysis for the benefit of non-specialists. The procedure is apparently fairly recently described. It seems to model variability of different regions in the cryo-EM reconstructions. However, the manuscript seems to imply that the procedure generates a temporal series of frames describing ligand binding. It is not clear how this would be derived. It is also not clear what validation is available for this approach, either in general or for the analysis specifically of the Ste2 samples. Also, it seems that “variability” and “variance” are used interchangeably in the text, which seems confusing.

The 3D variability analysis (3DVA) algorithm analyses the conformational landscape of a protein molecule using a linear subspace model of 3D structures. It essentially helps visualize and analyse the flexibility within a 3D cryo-EM reconstruction. The 3D variability components represent eigenvectors of the 3D covariance of images. To avoid confusion, we have replaced instances where “variance” was used with “variability”. The 3D variability analysis procedure indeed does not generate a temporal series of frames. We used the 3D variability analysis to gain information about the flexible engagement of different regions of the α -factor peptide with the receptor. In the Ste2•Ant state structure, the receptor transitions from density maps that lack signal for the N-terminus of the antagonist peptide to maps where this region becomes discernible, whereas the C terminus of the peptide is clearly visible. We can only state that the stable interactions made by the C-terminus of the peptide is consistent with previous kinetic data showing that the C-terminus binding domain of the α -factor engages first with the receptor, followed by the N-terminal signalling domain. It cannot be stated from the 3D variability analysis alone which region of the peptide engages first with the receptor. The 3D variability analysis procedure implemented in cryoSPARC has been validated with synthetic data (reference 10 in the manuscript) and has been routinely used without further validation for many cryo-EM datasets, including for GPCRs such as Adrenomedullin receptors AM1 and AM2 (*ACS pharmacology & translational science*, (2020) **3**(2), 263), GLP-1R (*Cell Reports* (2021), **36**(2), 109374) and CGRPR (*Science* (2021) **372**(6538)). We have also provided additional explanation for the 3D-variability analysis for better clarity.

We have clarified the text to read (lines 116-126):

'To interrogate the early events of peptide binding that lead to receptor activation, we performed 3D variability analysis of the cryo-EM data (Extended Data Fig. 6a-d). The 3D variability analysis algorithm¹⁰ provides detailed visualization of the flexibility within a 3D reconstruction by analysing the conformational landscape of a protein molecule using a linear subspace model of 3D structures. Analyses of 3D data along the vectors of motion revealed key facets of peptide binding that were not evident in the consensus cryo-EM maps. In the Ste2•Ant state (Extended Data Fig. 6b), the receptor transitions from density maps that lack signal for the N-terminus of the antagonist peptide to maps where this region becomes discernible, whereas the C terminus of the peptide is clearly visible. The stable engagement of the C-terminal region with the receptor is consistent with previous kinetic data suggesting that the C-terminus of the peptide engages with the receptor first¹¹.'

In Extended Data Fig. 6, it would seem to be important to indicate exactly what structures the displayed peptide chains, as opposed to the surface representations, correspond to. Is one the static consensus structure? The two representations are clearly not from the same structures, since they don't overlap.

We have added the following text to the legend for Extended Data Fig. 6 (lines 871-874):

'The displayed density maps (surface representations) represent the indicated frames obtained from the 3D variability analysis procedure. The atomic model is from the respective cryo-EM structures that has been docked into the map frames for better visualization.'

Why are different PC components displayed for the different samples? Why is there no PC1 from the antagonist sample? How was it decided how many PC's to display in each case?

Since we observed similar results from all the principal components, we chose one or two PC's for clear visualization. We have shown results from all the principal components (normal modes) in Supplementary Videos 1-4.

9. Extended Data Fig. 4 shows modeling only of the helical segments in the structures. Given the importance of loop regions, and the apparent poorer fit to the data at the helix extremities, it would seem to be important to provide a similar Extended Data figure for the loops. The poorer modeling does not necessarily seem to correlate with regions of lower resolution shown in Extended Data Fig. 3.

Extended Data Fig. 4e has been added to show densities from the loop regions as well. Extended Data Fig. 3 (local resolutions) has been modified to show a narrower resolution range to highlight the variation across resolution.

10. If C2 symmetry was applied to the reconstructions, why are the ligand binding pocket volumes of the two monomers different for each structure shown in Extended Data Fig. 5e?

Although C2 symmetry was applied during the last stages of 3D cryo-EM map reconstruction, we did not apply symmetry constraints during atomic model refinements leading to slight differences between the two protomer chains. We have now applied symmetry constraints during model refinements, and the binding pocket volumes calculated from these models show similar volumes for both the ligand binding pockets (see Extended Data Fig. 5e). Note that Ste2•Ag•G was refined in C1 symmetry resulting in slight differences in the calculated binding pocket volumes.

11. It is stated in the text that, "The flattening of the RMSD curves suggest (sic) that all simulations are equilibrated. However, such flattening with simulation time does not seem apparent in Extended Data Fig. 5a. If there is not flattening, what does this imply about the simulations?"

Flattening of the RMSD curve indicates that the protein structural dynamics has equilibrated at a given temperature. We have now revised the figure showing the moving average of the RMSD to clearly show the flattening in the change in the RMSD especially towards the last 100ns. We have calculated the fluctuation in the moving average for each system which is shown for each curve in Extended data Fig. 7a. The fluctuation in RMSD ranges from 0.03 to 0.2Å demonstrating the flattening of the RMSD.

12. There are many reports that Ste2p can activate mammalian or yeast-mammalian chimeras. How does this fit with the proposed specific mechanism of class D receptor activation?

It had previously been shown that the pheromone response in yeast could be coupled to Gpa1/mammalian G protein chimeras with different levels of efficacy and potency (*Yeast* 16.1 (2000): 11-22). This would seem reasonable due to the high sequence conservation among the mammalian G α subunits and Gpa1 (Human Go and *S. cerevisiae* Gpa1 share amino acid sequence identity and similarity of 51% and 67%, respectively), and chimeras with higher similarity to Gpa1 led to better EC₅₀ values for pheromone response in this study. Although the rearrangements within helices 5, 6 and 7

required for activation are different in the Ste2 dimer compared to the known activation mechanism of mammalian GPCR monomers, once the active-like state of Ste2 is reached and the G protein binding site becomes available, chimeric G α subunits may also be able to make productive interactions.

13. The manuscript seems to rely heavily on molecular dynamics calculations to address points that may be directly addressable from the structural data. Given the quality of the structural data showing changes in dimer contacts in different states, it seems gratuitous to then focus on residues that remain in contact 60% of the time in MD simulations. A similar argument could be made for allosteric changes in the receptor, given that the particular changes in the different states can be directly observed.

The structures we determined are snapshots along the activation pathway of Ste2 and represent a thermodynamically stable state that is amenable to analysis by structural biology. The MD simulations give information on the dynamics of the receptor structures and gives an idea on the percentage of time during a simulation that specific interactions may exist. This is a reflection on their relative stabilities and therefore the importance of the interactions in the structure. Thus, structural biology and MD simulations give different information that complements each other.

Referee #3 (Remarks to the Author):

1. The name of the force field used and the ionization state of His residue found in the α -factor should be given.

Sorry that we missed mentioning it in the Methods section. We used the CHARMM36m forcefield and the following text has been added (lines 705-706):

'All-atom molecular dynamics (MD) simulations were performed using the CHARMM36m forcefield⁶⁴...'

The His residue in the α factor was neutral with the proton on the epsilon nitrogen as determined by their binding environment and we have added the following text (lines 710-711):

'Histidine residues in the ligands were in the uncharged imidazole form.'

2. In Extended Figure 7b, allosteric interaction networks are given for different systems. It can be understood that one can get different pathways for antagonist-bound receptor; however, it is interesting to observe that pathways differ in the G protein-bound system. It would be informative if the authors can comment on it.

Multiple studies on class A GPCRs have shown that the binding of the G protein to the receptor enhances the binding affinity of agonists and reduces the affinity of the antagonists (doi: 10.1126/science.aau5595, doi: 10.1016/j.str.2018.12.007, doi: 10.1038/nature18324). The active-like system is bound to the agonist alone (no G protein) whilst the active state has both the agonist and the G protein bound. Therefore, we expect that the allosteric pipelines would be different due to the presence of the G proteins. We have clarified the text (lines 234-237) to read:

'G protein coupling affected allosteric communication within the Ste2 dimer, as seen by the differences between Ste2•Ag and Ste2•Ag•G, analogous to structural changes at the orthosteric binding site and extracellular surface observed upon coupling to a G protein¹².'

Reviewer Reports on the First Revision:

Referee #1 (Remarks to the Author):

Overall, the authors have done a fine job addressing my comments and suggestions. Thank you.

The only remaining issue is in regards to the sentence "In addition, Class C GPCR dimers are asymmetric and couple to a single G protein³³, in contrast to the symmetric dimer of Ste2 that can couple simultaneously to two G proteins¹."

Reference 33 is on GABAB-Gi. However, GABAB is a heterodimer where GABA engages the GABAB1 subunit and G protein engages the GABAB2 receptor subunit. It is thus better to reference a Family C homodimer for the noted asymmetry that promotes G coupling to one subunit (see mGlu2-Gi, PDB:7MTS).

Also, a couple of references seem to be repeated in the Ref list.

Author Rebuttals to First Revision:

I have addressed the referee's final comments by adding the requested reference and deleting duplicated references